# Reverse-Complement Equivariant Networks for DNA Sequences

**Vincent Mallet**
Structural Bioinformatics Unit, Department of Structural Biology and Chemistry,
Institut Pasteur, CNRS UMR3528, C3BI, USR3756
Mines ParisTech, PSL University, Center for Computational Biology
vincent.mallet96@gmail.com

**Jean-Philippe Vert**
Google Research, Brain team, Paris
jpvert@google.com

## Abstract

As DNA sequencing technologies keep improving in scale and cost, there is a growing need to develop machine learning models to analyze DNA sequences, e.g., to decipher regulatory signals from DNA fragments bound by a particular protein of interest. As a double helix made of two complementary strands, a DNA fragment can be sequenced as two equivalent, so-called *Reverse Complement* (RC) sequences of nucleotides. To take into account this inherent symmetry of the data in machine learning models can facilitate learning. In this sense, several authors have recently proposed particular RC-equivariant convolutional neural networks (CNNs). However, it remains unknown whether other RC-equivariant architectures exist, which could potentially increase the set of basic models adapted to DNA sequences for practitioners. Here, we close this gap by characterizing the set of all linear RC-equivariant layers, and show in particular that new architectures exist beyond the ones already explored. We further discuss RC-equivariant pointwise nonlinearities adapted to different architectures, as well as RC-equivariant embeddings of $k$-mers as an alternative to one-hot encoding of nucleotides. We show experimentally that the new architectures can outperform existing ones.

## 1 Introduction

Incorporating prior knowledge about the structure of data in the architecture of neural networks is a promising approach to design expressive models with good generalization properties. In particular, exploiting natural symmetries in the data can lead to models with fewer parameters to estimate than agnostic approaches. This is especially beneficial when the amount of available data is limited. A famous example of such an architecture is the convolutional neural network (CNN) for 1D sequences or 2D images, which is well adapted to problems which are invariant to translations in the data, while exploiting multiscale and local information in the signals. Motivated by the success of CNNs, there has been a fast-growing body of research in recent years to build the theoretical underpinnings and design architectures and efficient algorithms to systematically exploit symmetries and structures in the data [3].

A central idea that has emerged is to formalize the symmetries in data by a particular *group action* (e.g., the group of translations or rotations on images), and to create multilayer neural networks which, by design, "behave well" under the action of the group. This is captured formally by the concept of *equivariance*, which states that each equivariant layer should be designed to be subject to the group

35th Conference on Neural Information Processing Systems (NeurIPS 2021).

action (e.g., we should be able to "translate" or "rotate" the signal in each layer), and that when an input data is transformed by a particular group element, then its representation in an equivariant layer should also be transformed according to the same group element. While it is easy to see that convolutional layers in CNNs are equivariant to translations, Cohen and Welling [7] formalized the concept of group equivariance CNN (G-CNN) for more general groups and showed in particular how to design convolutional layers equivariant not only to translations but also to reflections and to a discrete set of rotations. Following this seminal work, the theoretical foundations of group equivariant neural networks were then expanded, going beyond regular representations [9], for more groups [2, 18, 37, 40], in less regular spaces [8, 10] or with more general results on their generality and universality [11, 13, 14, 22]. The main applications were developed with the groups of rotations in 2D and 3D, mostly to computer vision problems, but also in biology with histopathology [17, 23], medicine [41] and quantum chemistry [33].

In this paper, we explore and study the potential benefits of equivariant architectures for an important class of data, namely deoxyribonucleic acid (DNA) sequences. DNA is the major form of genetic material in most organisms, from bacteria to mammals, which encodes in particular all proteins that a cell can produce and which is transmitted from generation to generation. The study of DNA in humans and various organisms has led to tremendous progress in biology and medicine since the 1970s, when the first DNA sequencing technologies were invented, and the collapsing cost of sequencing in the last twenty years has accelerated the production of DNA sequences: there are for example about 2.8 billion sequences for a total length of $\sim 10^{13}$ nucleotides publicly available at the European Nucleotide Archive (ENA[1]). Unsurprisingly in such a data-rich field, machine learning-based approaches are increasingly used to analyze DNA sequences, e.g., in metagenomics to automatically predict the species present in an environment from randomly sequenced DNA fragments [26, 28, 36, 38] and to detect the presence of viral DNA in human samples [36], in functional genomics to predict the presence of protein binding sites or other regulatory elements in DNA sequences of interest [16, 24, 30, 35, 43, 44], to predict epigenetic modifications [25], or to predict the effect of variations in the DNA sequence on a phenotype of interest [1, 46].

Due to the sequential nature of DNA and the translation-equivariant nature of the questions addressed, many of these works are based on 1D CNN architectures, although recently transformer-based language models have also shown promising results on various tasks [6, 20, 42]. However, besides translation, DNA has an additional fundamental symmetry that has been largely ignored so far: the so-called reverse complement (RC) symmetry, due to the fact that DNA is made of two strands oriented in opposite direction and encoding complementary nucleotides. In other words, a given DNA segment can be sequenced as two RC DNA sequences, depending on which strand is sequenced; any predictive model for, e.g., DNA sequence classification should therefore be RC-invariant, which calls for RC-equivariant architectures. While strategies based on data augmentation and prediction averaging has been commonly used to handle the need for RC invariance [1, 32], one translation- and RC-equivariant CNN architecture has been proposed and led to promising results [4, 29, 34]. However, it remains unclear whether that architecture is the only one that can encode translation- and RC-equivariance, or if alternative models exist to complement the toolbox of users wishing to develop deep learning models for DNA sequences.

Using the general theory of equivariant representations, in particular steerable CNNs [9], we answer that question by characterizing the set of all linear translation- and RC-equivariant layers. We show in particular that new architectures exist beyond the ones already explored by [4, 29, 34], which in the language of equivariant CNNs only make use of the regular representation [7] while more general representations lead to different layers. We further discuss RC-equivariant pointwise nonlinearities adapted to different representations, as well as RC-equivariant embeddings of $k$-mers as an alternative to one-hot encoding of nucleotides. We test the new architecture on several protein binding prediction problems, and show experimentally that the new models can outperform existing ones, confirming the potential benefit of exploring the full set of RC-equivariant layers when manipulating DNA sequences with deep neural networks.

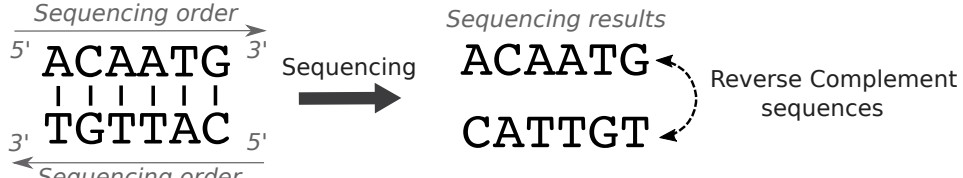

Figure 1: Illustration of the reverse-complement symmetry. Both DNA strands get sequenced in opposite directions resulting in redundant information.

## 2 Methods

### 2.1 Group action of translation and reverse complementarity on DNA sequence

DNA is a long polymer made of two intertwined strands, forming the well-known double-helical structure. Each strand is a non-symmetric polymer that can be described as an oriented chain of four possible monomers called nucleotides and denoted respectively $\{A, C, G, T\}$. The two strands are oriented in opposite directions, and their nucleotides face each other to form hydrogen bonds. They interact at each position in a deterministic way because only two nucleotides pairings can happen: $(A,T)$ and $(G,C)$. Thus, given a nucleotide sequence on one strand, we can deduce the so-called RC sequence of its corresponding strand by complementing each nucleotide and reversing the order (**Figure 1**). When a double-stranded DNA fragment is sequenced, the two strands are first separated and, typically, only one of them is randomly selected and is decrypted by the machine. This implies that any given DNA fragment can be equivalently described by two RC sequences of nucleotides. Moreover, several genomic learning tasks amount to a sequence annotation that does not depend on the strand. For example, a protein can bind a double-stranded DNA fragment, and both strands of the bound part can get sequenced. This motivates the search for equivariance to this RC-action for the prediction functions. Moreover, the sequencing often results in long sequences where the relevant parts of the sequence do not correlate with their position. The task of prediction over genomic sequences is thus largely translation equivariant, which explains why the community settled on the use of CNNs to train and predict on arbitrary length segments.

To formalize mathematically the translation and RC operations on DNA sequences, we first encode the raw genetic sequence as a signal function in $F_0 = \left\{ f : \mathbb{Z} \to \{0, 1\}^4 \right\}$, as the one hot encoding of the nucleotide content for each integer position. Because of the finite length of this polymer, we assume that beyond a compact support this function takes a constant value of zero. The group $(\mathbb{Z}, +)$ of translations acts naturally on this encoding by $T_u(f)(x) = f(x - u)$, for a translation $u \in \mathbb{Z}$, and the RC operations amounts to the following : $RC(f)(x) = \sigma(-1)[f(-x)]$, where $\sigma(-1)$ is the $4 \times 4$ permutation matrix that exchanges complementary bases $A/T$ and $C/G$ (while we denote by $\sigma(1)$ the $4 \times 4$ identity matrix). We notice that $RC$ is a linear operation on $F_0$ that satisfies $RC^2 = I$, and thus that the RC operation is a group representation on $F_0$ for the group $\mathbb{Z}_2 = \{1, -1\}$ endowed with multiplication.

To jointly consider translations and RC actions, we naturally consider the semi-direct product group $G = \mathbb{Z} \rtimes \mathbb{Z}_2$. Elements $g \in G$ can be written as $g = ts$ with $t \in \mathbb{Z}, s \in \mathbb{Z}_2$ and the group $G$ acts on $F_0$ by the action $\pi_0$ defined by:

$$\forall ts \in G, \quad \forall (f, x) \in F_0 \times \mathbb{Z}, \quad \big(\pi_0(ts)f\big)(x) = \sigma(s)[f(s(x - t))].$$

In other words, $\pi_0$ is the representation of $G$ on $F_0$ induced by the representation $\sigma$ of RC on $\mathbb{R}^4$ [9].

### 2.2 Features spaces of equivariant layers

Let us now describe the structure of intermediate layers of a neural network equivariant to translations and RC. Following the theory of steerable CNNs [9], we consider successive representations of the input DNA sequence in the following way:

**Definition 1.** *Given $\rho$ a representation of $\mathbb{Z}_2$ on $\mathbb{R}^D$ for some $D \in \mathbb{N}^*$, a $\rho$-feature space is the set of signals $F = \{f : \mathbb{Z} \to \mathbb{R}^D\}$ endowed with the $G$ group action $\pi$, known as the representation*

---

[1]As of May, 2021: https://www.ebi.ac.uk/ena

*induced by $\rho$ :*

$$\forall ts \in G, \quad \forall (f, x) \in F \times \mathbb{Z}, \quad \big(\pi(ts)f\big)(x) = \rho(s)[f(s(x - t))]. \tag{1}$$

With this definition, we see in particular that the one-hot encoding input layer maps the input DNA sequence to a $\sigma$-feature space, and that the dimension (i.e., number of channels in the language of deep learning) and group action of $\rho$-feature space are fully characterized by the representation $\rho$. Interestingly, the theory of linear group representations allows us to characterize more precisely *all* such representations:

**Theorem 1.** *For any representation $\rho$ of $\mathbb{Z}_2$ on $\mathbb{R}^D$, there exist $a, b \in \mathbb{N}$ such that $a + b = D$ and an invertible matrix $P \in GL(\mathbb{R}^D)$ such that*

$$\forall s \in \mathbb{Z}_2, \quad \rho(s) = P \, Diag(I_a, sI_b)P^{-1}.$$

In other words, combining Definition 1 and Theorem 1, we see that any $\rho$-feature space that we will use to build translation- and RC-equivariant layers is fully characterized by a triplet $(P, a, b)$, which we call its *type*, and which characterizes both its dimension $D = a + b$ and the action of the group $G$ by (1). By slight abuse of language, we also refer to $(P, a, b)$ as the type of $\rho$.

Theorem 1 is a standard result of group theory, which explicits the decomposition of any representation $\rho$ in terms of so-called irreducible representation, or *irreps*. In the case of $\mathbb{Z}_2$, there are exactly two irreps which act on $\mathbb{R}$, namely, $\rho_1(s) = 1$ and $\rho_{-1}(s) = s$. If $\rho$ has type $(P, a, b)$, then it means that it can be decomposed as $a$ times $\rho_1(s)$ and $b$ times $\rho_{-1}(s)$. In the particular case where $P$ is the identity matrix, i.e., when we consider a type $(I, a, b)$, then $\rho(s)$ is a diagonal matrix for any $s \in \mathbb{Z}_2$, and each channel of $F$ is acted upon by a single irrep. In that case, we will call the channels of type "1" (resp. "-1") if they are acted upon by $\rho_1$ (resp. $\rho_{-1}$), and we will say that $F$ is an "irrep feature space".

Now, let us introduce another special case. Since $\mathbb{Z}_2$ is finite of cardinality 2, let us consider the *regular representation* $\rho_{reg}$ of $\mathbb{Z}_2$ on $\mathbb{R}^2$ defined by:

$$\rho_{reg}(1) = \begin{pmatrix} 1 & 0 \\ 0 & 1 \end{pmatrix}, \quad \rho_{reg}(-1) = \begin{pmatrix} 0 & 1 \\ 1 & 0 \end{pmatrix}.$$

One can easily check that $\rho_{reg}$ is of type $(P_{reg}, 1, 1)$, where $P_{reg} = \begin{pmatrix} 1 & 1 \\ 1 & -1 \end{pmatrix}$. It corresponds to a $\rho$-feature space with two channels, where the RC operations flips the two channels (and of course the sequence coordinates).

Let us now consider feature spaces of interest. In the input layer, nucleotides are one-hot encoded in a certain order, let us say $(A, T, G, C)$. As stated above, this input space is acted upon by $\sigma$, a $2-$cycle that swaps bases $A/T$ and $C/G$. We see that we can rewrite $\sigma = (\rho_{reg} \oplus \rho_{reg}) := (\rho_{reg}^{\oplus 2})$, where $\oplus$ is the bloc-diagonal operation. Because $\rho_{reg}$ is of type $(P_{reg}, 1, 1)$, we can diagonalize $\sigma$ with $(P_{reg}^{\oplus 2})$ and the diagonal would be alternated +1 and -1 values. Thus, there exists a permutation $\Pi$ such that $\sigma$ is of type $(P, 2, 2)$, with $P = \Pi(P_{reg}^{\oplus 2})\Pi^{-1}$. These concepts are illustrated in Supplementary Section A.1

Interestingly, all RC-equivariant layers proposed so far in [4, 29, 34] follow a similar pattern: the channels go by pair, and the RC action amounts to flipping the channel values within a pair and reversing the sequence coordinates. In our formalism, this corresponds to channels of type $(P, a, a)$, where $a \in \mathbb{N}^*$ is the number of pairs of channels, and where up to a permutation of channels the matrix $P$ satisfies $P = \Pi(P_{reg}^{\oplus a})\Pi^{-1}$. Following [34], we will refer to these layers as *Reverse Complement Parameter Sharing* (RCPS) layers below.

This highlights the fact that translation- and RC-equivariant layers explored so far are equivariant according to Definition 1, but that there exists potentially many other equivariant layers, obtained in particular by allowing $\rho$-feature spaces of types $(P, a, b)$ where $a \neq b$, on the one hand, and where $P$ is not a direct sum of $P_{reg}$, on the other hand. We investigate such variants below.

## 2.3 Equivariant linear layers

While Definition 1 characterizes $\rho$-feature space in terms of structure and group action, an equivariant multilayer neural network is built by stacking $\rho$-feature spaces on top of each other and connecting

them with equivariant layers. Cohen et al. [11, Theorem 2] gives us a general result about such equivariant mappings. Here, we apply this result to our specific data and group, and characterize the class of equivariant linear layers, i.e., the linear functions $\phi : F_n \to F_{n+1}$ that satisfy $\pi_{n+1}\phi = \phi\pi_n$, where $\pi_n$ and $\pi_{n+1}$ are respectively the group action on $F_n$ and $F_{n+1}$.

**Theorem 2.** *Given two representations $\rho_n$ and $\rho_{n+1}$ of $\mathbb{Z}_2$, of respective types $(P_n, a_n, b_n)$ and $(P_{n+1}, a_{n+1}, b_{n+1})$ with $a_n + b_n = D_n$ and $a_{n+1} + b_{n+1} = D_{n+1}$, and respective $\rho_n$- and $\rho_{n+1}$-feature spaces $F_n$ and $F_{n+1}$, a linear map $\phi : F_n \to F_{n+1}$ is equivariant if and only if it can be written as a convolution:*

$$\forall (f, x) \in F_n \times \mathbb{Z}, \quad \phi(f)(x) = \sum_{y \in \mathbb{Z}} \kappa(y - x) f(y), \tag{2}$$

*where the kernel $\kappa : \mathbb{Z} \to \mathbb{R}^{D_{n+1} \times D_n}$ satisfies:*

$$\forall x \in \mathbb{Z}, \quad \kappa(-x) = \rho_{n+1}(-1)\kappa(x)\rho_n(-1), \tag{3}$$

*or equivalently:*

$$\forall x \in \mathbb{Z}, \quad \kappa(x) = P_{n+1} \begin{pmatrix} \alpha(x) & \beta(x) \\ \gamma(x) & \delta(x) \end{pmatrix} P_n^{-1}, \tag{4}$$

*where $\alpha : \mathbb{Z} \to \mathbb{R}^{a_{n+1} \times a_n}$ and $\delta : \mathbb{Z} \to \mathbb{R}^{b_{n+1} \times b_n}$ are even, while $\beta : \mathbb{Z} \to \mathbb{R}^{a_{n+1} \times b_n}$ and $\gamma : \mathbb{Z} \to \mathbb{R}^{b_{n+1} \times a_n}$ are odd functions.*

As stated in Cohen et al. [11], "Convolution is all you need" to define linear layers which are equivariant to our group. In addition, Theorem 2 characterizes all the convolution kernels that ensure equivariance through the two equivalent constraints (3) and (4).

To illustrate this result, let us consider two RCPS feature spaces $F_n$ and $F_{n+1}$ of respective types $(\Pi_n(P_{reg}^{\oplus a_n})\Pi_n^{-1}, a_n, a_n)$ and $(\Pi_{n+1}(P_{reg}^{\oplus a_{n+1}})\Pi_{n+1}^{-1}, a_{n+1}, a_{n+1})$. Then, the channels in $F_n$ and $F_{n+1}$ go by pair, and if we consider a slice $\tilde{\kappa} : \mathbb{Z} \to \mathbb{R}^{2 \times 2}$ of the convolution kernel $\kappa$ describing how a pair of channels in $F_n$ maps to a pair of channels in $F_{n+1}$, (3) gives the constraint:

$$\tilde{\kappa}(-x) := \begin{pmatrix} \tilde{\kappa}_{11}(-x) & \tilde{\kappa}_{12}(-x) \\ \tilde{\kappa}_{21}(-x) & \tilde{\kappa}_{22}(-x) \end{pmatrix} = \begin{pmatrix} 0 & 1 \\ 1 & 0 \end{pmatrix} \tilde{\kappa}(x) \begin{pmatrix} 0 & 1 \\ 1 & 0 \end{pmatrix} = \begin{pmatrix} \tilde{\kappa}_{22}(x) & \tilde{\kappa}_{21}(x) \\ \tilde{\kappa}_{12}(x) & \tilde{\kappa}_{11}(x) \end{pmatrix}.$$

We recover exactly the constraints of the RCPS filters first proposed by [34], proving as a consequence of Theorem 2 that RCPS convolution filters describe exactly *all* equivariant linear mappings between RCPS feature spaces.

Moreover, if we now consider any two feature spaces $F_n$ and $F_{n+1}$ of respective types $(P_n, a_n, b_n)$ and $(P_{n+1}, a_{n+1}, b_{n+1})$, then Equation (4) tells us that up to multiplications by matrices $P_{n+1}$ and $P_n^{-1}$, the kernel is expressed in terms of even and odd functions, which can be trivially implemented with parameter sharing. For example, to represent the even function $\alpha$, one just need to parameterize the values of $\alpha(x)$ for $x \geq 0$, and complete the negative values by parameter sharing $\alpha(-x) = \alpha(x)$. Hence, the parameter sharing idea used in RCPS [34] extends to any equivariant linear map.

Instead of using (4) to parameterize equivariant convolution kernels, one may also directly write the constraints (3) for specific representations, and potentially save the need of multiplication by $P_{n+1}$ and $P_n^{-1}$ in (4). This is for example the case in RCPS layers [34], and more generally for channels acted upon by the regular representation; for the sake of completeness, we derive in Appendix A.4 the constraints to go from and to the regular representation or the irreps, and use them in our implementation.

## 2.4 Equivariant nonlinear layers

Besides equivariant linear layers, a crucial component needed for multilayer neural networks is the possibility to have equivariant nonlinear layers, such as nonlinear pointwise activation functions or batch normalization [19]. In this section, we discuss particular nonlinearities that are adapted to various equivariant layers.

**Pointwise activations.** Let us begin with pointwise transformations, that encompass most activation functions used in deep learning. Pointwise transformations are formally defined as follows: given

a function $\theta : \mathbb{R} \to \mathbb{R}$ and a vector space $V = \mathbb{R}^A$ for some index set $A$, the pointwise extension of $\theta$ to $V$ is the mapping $\bar{\theta}_V : V \to V$ defined by $\bar{\theta}_V(f)(a) = \theta(f(a))$, for any $(f, a) \in V \times A$. For a $D$-dimensional representation $\rho$ of $\mathbb{Z}_2$ and a $\rho$-feature space $F$ with $G$-group action $\pi$, we say that a pointwise extension $\bar{\theta}_F : F \to F$ is equivariant if it commutes with $\pi$, i.e., $\pi\bar{\theta}_F = \bar{\theta}_F\pi$. By definition of the group action (1), this is equivalent to saying that the pointwise extension $\bar{\theta}_{\mathbb{R}^D}$ of $\theta$ to $\mathbb{R}^D$ commutes with $\rho$. The following theorem gives an exhaustive characterization of a large class of equivariant pointwise extensions for any $\rho$-feature space:

**Theorem 3.** *Let $\rho$ be a representation of $\mathbb{Z}_2$ and $\theta : \mathbb{R} \to \mathbb{R}$ be a continuous function with left and right derivatives at $0$. Let $F$ be a $\rho$-layer and $\bar{\theta}_F : F \to F$ be the point-wise extension of $\theta$ on this layer. Then $\bar{\theta}_F$ is equivariant if and only if at least one of the following cases holds:*

1. *$\theta$ is a linear function.*

2. *$\theta$ is an affine function, and $\rho(-1)\mathbf{1} = \mathbf{1}$.*

3. *$\theta$ is not an affine function, and there exists a permutation matrix $\Pi$, integers $a, b, c, d \in \mathbb{N}$, and scalars $(\lambda_1, \ldots, \lambda_a) \in (\mathbb{R}_+^*)^a$, such that $\rho$ decomposes as*

$$\Pi^{-1}\rho(-1)\Pi = \bigoplus_{i=1}^{a} \begin{pmatrix} 0 & \lambda_i \\ \lambda_i^{-1} & 0 \end{pmatrix} \oplus \begin{pmatrix} 0 & -1 \\ -1 & 0 \end{pmatrix}^{\oplus b} \oplus (1)^{\oplus c} \oplus (-1)^{\oplus d}. \quad (5)$$

   *In that case,*

   - *Either $b = d = 0$ and $\forall i, \lambda_i = 1$ and $\theta$ is any function,*
   - *Or $b = d = 0$ and $\exists i, \lambda_i \neq 1$ and $\theta$ is a leaky ReLu function.[2]*
   - *Or $b + d > 0$ and $\forall i, \lambda_i = 1$ and $\theta$ is an odd function,*

The first case in Theorem 3 is of little interest, since pointwise linear functions are always equivariant to linear group actions. The second case essentially says that adding a constant to a pointwise linear function is only equivariant for representations $\rho$ such that the sum of all rows of $\rho(-1)$ is equal to $1$. This holds for example for the regular representation and the RCPS layers, but not for an irrep feature space of type $(I, a, b)$ with $b > 0$, since in that case, some rows have a single "-1" entry. The most interesting case is the third one, since it describes what pointwise nonlinearities one can apply. The condition (5) on the decomposition of $\rho$ essentially excludes all representations that have more than one nonzero value in at least one row of $\rho(-1)$. Among valid $\rho$'s that decompose as (5), we see that the regular representation (corresponding to the first block in (5) with $\lambda_i = 1$)), used in RCPS, stands out as the only that allows *any* nonlinearity, besides of course invariant channels of type "+1" (third block in (5)). Replacing a "1" in the regular representation by a scalar $\lambda_i \neq 1$ (in the first block of (5), with $b = d = 0$) creates a valid representation $\rho$, however only leaky ReLu pointwise nonlinearities can be applied in that case. Another case of practical interest is the irrep feature space of type $(I, c, d)$ for some $c > 0$ and $d > 0$. By Theorem 3, only odd nonlinearities are allowed in that case, such as the hyperbolic tangent function. Finally, one should keep in mind that other representations, which do not satisfy the conditions listed in Theorem 3, do not allow any equivariant nonlinear pointwise transform; this is for example the case of $\rho(-1) = \begin{pmatrix} 0 & -1/2 \\ -2 & 0 \end{pmatrix}$, which is a valid representation of $\mathbb{Z}_2$ but does neither meet the condition to accept affine activations (case 2), nor to accept nonlinear activations (case 3) because $\rho(-1)$ does not decompose according to (5).

**Other activation functions** Besides pointwise transformations from a $\rho$-feature space to itself characterized in Theorem 3, the set of nonlinear equivariant layer is tremendous and the design choices are endless. A first extension is to keep pointwise activation, but to allow different nonlinearities on different channels, e.g., by using any function on the "+1" channels and an odd function on the "-1" channels of an irrep feature space. Another relaxation is to use different input and output representations. While odd functions will not affect the field type, even functions will turn a field of type "-1" into a "+1" type. It is well known that any function decomposes into a sum of an odd and even function. Therefore, given $\rho$, a representation decomposed as in (5), any point-wise non-linearity can be used in a $\rho$-feature space by first decomposing it into its odd and even components and applying each component separately for the second and fourth blocks.

---

[2]A leaky ReLu function is $\theta(x) = \alpha_{sign(x)}x$ for some $(\alpha_+, \alpha_-) \in \mathbb{R}^2$.

Other possibilities exist and include creating new representations by tensorization, which amounts to taking pointwise products between different channels [13, 21, 37]. or using non point-wise activation layers, that act on several coupled dimensions, such as the ones used in [37]. For instance, we could apply the max function to paired channels. These possibilities are discussed in [39]

**Batch normalization**    An equivariant batch normalization was introduced by [34]. It considers a feature map and its reverse complement as two instances, which is easy to do because the reverse complement feature map is already computed when using regular representation. We propose another batch normalization for irrep feature spaces that also gives the result we would have had if the batch contained all the reverse complement of its sequences. For the "+1" dimensions, it amounts to scaling as we would have the same values twice. For the "-1" dimensions, we enforce a zero mean and compute a variance estimate based on this constraint.

**K-mers.**    Instead of the standard one-hot encoding of individual nucleotides as input layer, we propose to one-hot encode $k$-mers for $k \geq 1$, i.e., overlapping blocks of $k$ consecutive nucleotides. This technique is known to improve performance in several tasks [27, 28]. In order to implement it into an equivariant network, we need to know how the group acts on the $k$-mers space, made of $4^k$ elements. The simplest idea is to pair the index of the channels of two RC $k$-mers. Because some $k$-mers are their own reverse complement, the canonical way to do so is to have a representation that is a blend of "+1" irrep and regular representation. An alternative is to make the regular representation act on the $k$-mers instead by redundantly encoding these $k$-mers into paired dimensions. This is the strategy we follow in our implementation, to be more coherent with the usual input group action.

## 3   Experiments

We assess the performance of various equivariant architectures on a set of three binary prediction and four sequence prediction problems used by Zhou et al. [45] to assess the performance of RCPS networks. The binary classification problems aim to predict if a DNA sequence binds to three transcription factors (TFs), based on genome-wide binarized TF-ChIP-seq data for Max, Ctcf and Spi1 in the GM12878 lymphoblastoid cell-line [34]. The sequence prediction problems aim to predict TF binding at the base-pair resolution, using genome-wide ChIP-nexus profiles of four TFs-Oct4, Sox2, Nanog and Klf4 in mouse embryonic stem cells. For a more detailed explanation of the experimental setup, please refer to Zhou et al. [45]. We report "significant" differences in performance below when the P-value of a Wilcoxon signed rank test is smaller than $0.05$.

**Models.**    We build over the work of Zhou et al. [45] for both the binary and the sequence prediction problems. They benchmarked an equivariant RCPS architecture and a corresponding non-equivariant model, with the same number of filters and trained with data augmentation, which we respectively refer to as "RCPS" and "Standard" models below. The data augmentation scheme for the "Standard" model consists in adding to the training set the reverse complement sequences of all training sequences, which is a natural procedure to let the model "learn" the equivariance without encoding it explicitly in the architecture of the network. We checked empirically that data augmentation significantly improves the performance of non-equivariant models (Appendix A.6.1). In addition, we extend the RCPS architecture with one-hot encoding of $k$-mers as input layers, which we refer to as "Regular" below. Finally, we add to the comparison a new equivariant network where each RCPS layer is replaced by an $(I, a, b)$ layer with the same number of filters, which we call "Irrep" below. We also use $k$-mers and vary the ratio $a/(a + b)$ in this model. We combine the regular and "+1" dimensions with *ReLu* activations and the "-1" dimensions with a *tanh* activation.

**Influence of hyperparameters in equivariant models**    To assess the impact of different hyperparameters in the family of equivariant models we propose ($k$-mer length for Irrep and Regular, $a/(a+b)$ ratio for Irrep), we train equivariant models with different combinations of hyperparameters on the training set and assess their performance on the validation set, repeating the process ten times with different random seeds. We assess the performance of each run in terms of Area under the Receiver Operator Characteristic (AuROC), and show in Figure 2 the average performance reached by all runs with a given ratio $a/(a + b) \in \{0, 1/4, 1/2, 3/4, 1\}$ (left) and with a given $k \in \{1, 2, 3, 4\}$ (right). We see a clear asymmetry in the performance as a function of $a/(a + b)$, with poor performance when $a = 0$ and optimal performance for $a = 0.75$, significantly better than all other ratios tested.

This confirms that exploring different irreps may be valuable. As for the $k$-mer length, setting $k = 3$ gives the best performance and significantly outperforms all other values of $k$ tested. This confirms that going beyond one-hot encoding of nucleotides in equivariant architectures can be beneficial.

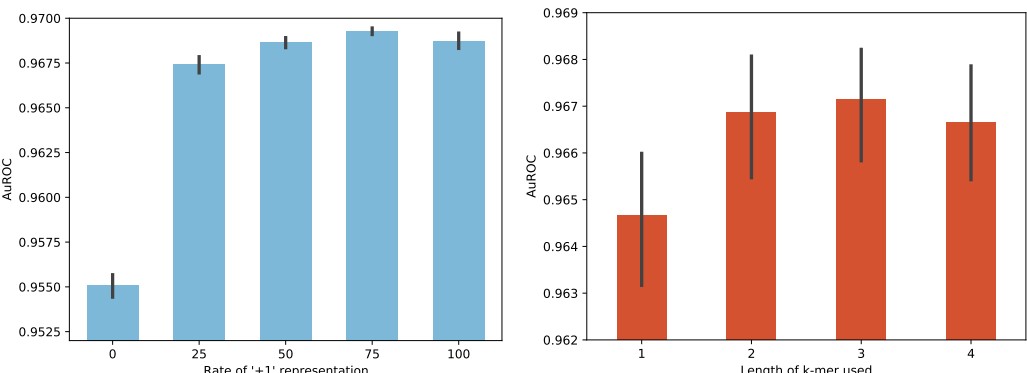

Figure 2: Average AuROC performance across four TFs and 10 random seeds for the Irrep model as a function of $a/(a + b)$ (*left*, also averaged over $k$ values) and for the Irrep and Regular models as a function of $k$ (*right*, also averaged over $a/(a + b)$ values for Irrep).

**Binary task.** We then compare the test set performance of three different models for the binary classification task: 1) Standard, 2) RCPS, and 3) the best Irrep or Regular equivariant model, where hyperparameters are selected based on the AuROC on the validation set, which we denote as "Best Equivariant". Figure 3 (left) shows the performance of each model on each TF task and overall. As already observed by [34], the equivariant RCPS architecture has a strong lead over the Standard, non-equivariant model in spite of data augmentation. Interestingly, we see that Best Equivariant is significantly better than RCPS on all tasks, and that the performance gain from RCPS to Best equivariant is of the same order as the performance gain from Standard to RCPS. This demonstrates that the family of equivariant architectures we introduce in this paper can lead to significant improvement over existing architectures.

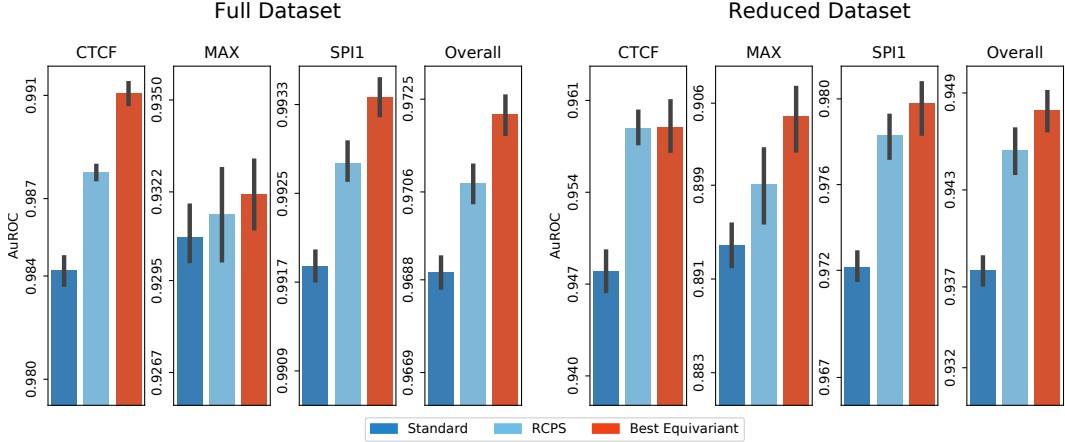

Figure 3: AuROC performance of the three different models (Standard, RCPS and Best equivariant after hyperparameter selection on the validation set) on the three binary classification problems CTCF, MAX and SPI1, as well as their average. Error bars correspond to an estimate of the standard error on 10 repeats with different random seeds. The left plot is the performance on the full datasets, while the right plot shows the performance where models are trained on a subset of 1,000 sequences only (notice the differences of AuROC values on the vertical axis in both plots).

**Reduced models.** Since equivariant architectures are meant to be particularly beneficial in the low-data regime [15], we further assess the performance of the three models on the same binary

classification problems but with only 1,000 sequences used to train the models, and show the results on Figure 3 (right). Overall, the performances are worse than in the full-data regime (Figure 3, left), which confirms that this is a regime where more data helps. We also see that the relative order of the three different methods remains overall the same, with Best Equivariant outperforming RCPS, which itself outperforms Standard. Interestingly, the gaps between the best and worse models widens in the low-data regime, showing that the prior is more useful in this setting. More precisely, there is a large gap of about $1\%$ between Best Equivariant and Standard in the low data regime, compared to a gap of about $0.3\%$ on the full dataset. We also investigated whether equivariant models converge faster to their solutions, but found not noticeable difference (Appendix A.6.2).

**On post-hoc models.** Zhou et al. [45] introduced the so-called *post-hoc* model, another equivariant method obtained by averaging the predictions of a Standard model over a sequence and its reverse-complement, and showed that it is competitive with and often outperforms RCPS. The post-hoc model only requires training and storing one network, but aggregates two predictions for each sequence at inference time. Because of that, the good performance of post-hoc may be due in part to the aggregation step common to all ensemble models [12]. To decipher the respective contributions of the network architecture, on the one hand, and of the aggregation of predictions, on the other hand, we add to the comparison an ensemble of two Standard models trained with different random seeds (*Ensemble Standard*) and an ensemble of two equivariant Irrep models (*Ensemble Irrep*) and present the results in Figure 4. We see that Ensemble Irrep strongly outperforms Best Equivariant, and both post-hoc and Ensemble Standard widely outperform the Standard architecture. This confirms that ensembling equivariant or non-equivariant models through post-hoc of ensemble aggregation is always useful (at the cost of increased computational time). We see that Ensemble Standard is not significantly different from post-hoc Standard on CTCF and SPI1, but that post-hoc Standard is better on MAX, suggesting that most of the benefits of post-hoc Standard indeed comes from the ensembling effect. Regarding the impact of the architecture for a given budget of predictions, we saw earlier than Best equivariant significantly outperforms Standard when a single prediction per test sequence is allowed, and see now that Ensemble Irrep strongly outperforms both post-hoc and Ensemble Standard when two predictions are allowed, thus confirming the benefit of equivariant architectures in all settings. We also see that a single Best equivariant models outperforms post-hoc and Ensemble Standard, indicating that enforcing equivariance throughout the network is not only faster but also more more accurate than averaging a non-equivariant model over group transformed inputs.

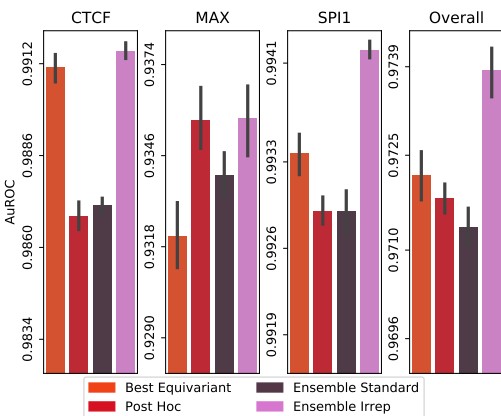

Figure 4: AuROC performance on the three binary classification problems, for the Best Equivariant model, the post-hoc Standard model, and an ensemble of two Standard or Irrep models. Error bars correspond to an estimate of the standard error on 10 repeats with different random seeds.

**Profile task.** We now compare the performance of different models on the profile prediction tasks. To limit the carbon footprint of this study, and based on the influence of hyperparameters on the binary task (Figure 2), we only test two equivariant models in addition to Standard and RCPS: a Regular model with $k = 3$, and an Irrep model with $k = 3$ and $a/(a + b) = 75\%$. We also assess the performance of post-hoc Standard (the best model in [45]), and an ensemble of two models of

the best performing equivariant model. Figure 5 shows the performance of all models in terms of Spearman correlation between the target profile and the predicted ones, on the full dataset (left) or a reduced experiment with only 1,000 training sequences (right).

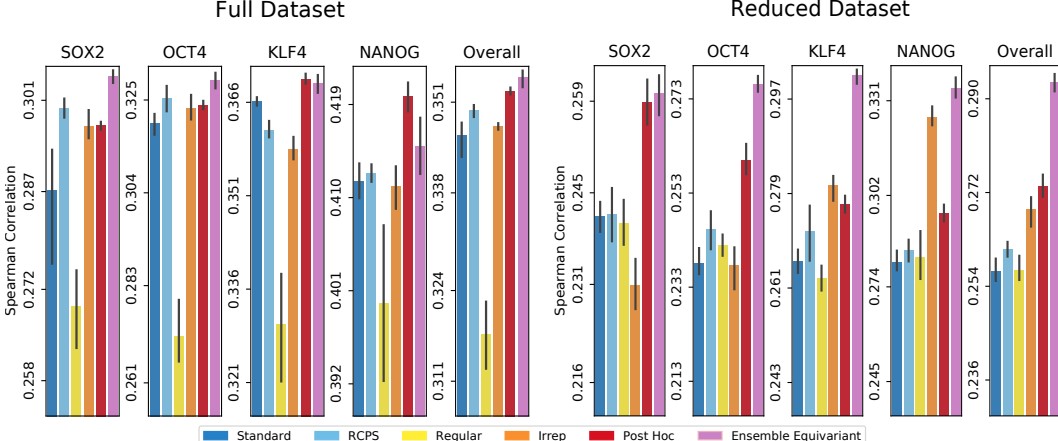

Figure 5: Spearman Correlation between true and predicted profiles by different methods for four data sets.

First of all, we observe as before that in the low-data regime, the gap between standard and equivariant networks grows in favor of equivariant ones. We also observe, surprisingly, that Irrep, which outperformed RCPS on the binary task, now underperforms it. A possible explanation could be that since this task aims to annotate an individual nucleotide, encoding the nucleotide level information using $k$-mers makes the signal blurry and decreases performance. However, in the reduced setting, Irrep performs better again. These results indicate that for now the best model should be chosen empirically on a validation set. Finally, despite good performance of post-hoc Standard, the ensemble equivariant model once again performs better for the same computational cost at inference.

**Experiment settings and computational cost.** All experiments were run on a single GPU (either a GTX1080 or a RTX6000), with 20 CPU cores. The binary classification experiments were shorter to train. To limit our carbon footprint, we chose to run more experiments on this task, e.g., for hyperparameter tuning and to reduce the number of replicates for the profile task. The total runtimes of each of those tasks were approximately of a week.

## 4 Conclusion

In this paper, we addressed the problem of including the RC symmetry prior in neural networks. Leveraging the framework of equivariant networks, in particular steerable CNNs, we deepened existing methods by unraveling the whole space of linear layers and pointwise nonlinearities that are translation and RC-equivariant. We also investigated the links between the linear representations and the non-linear layers of neural networks, exposing the special role of the regular representation in equivariant networks. Finally, we implemented new linear and nonlinear equivariant layers and make all these equivariant layers available in Keras [5] and Pytorch [31]. [3] We then explored empirically how this larger equivariant functional space behaves in terms of learning. Our best results improve the state of the art performance of equivariant networks, showing that new equivariant architectures can have practical benefits. In the future we plan to test more deeply the newly proposed architectures on prediction tasks involving double-stranded DNA, such as DNA-protein binding prediction, epigenetics or metagenomics. On the theoretical side, we characterized equivariant pointwise nonlinearities that preserve the layer type, but more general nonlinear transforms (e.g., not pointwise, or changing the layer type) remain to be fully characterized.

---

[3]code available at `https://github.com/Vincentx15/Equi-RC`

## Acknowledgments and Disclosure of Funding

V.M. is recipient of a doctoral fellowship from the INCEPTION project [PIA/ANR-16-CONV-0005] and benefits from support from the CRI through Ecole Doctorale FIRE - Programme Bettencourt. We thank Marie Dechelle, Jacques Boitreaud, Carlos G. Oliver and Guillaume Bouvier for reviewing the manuscript. We thank Avanti Shrikumar, Hannah Zhou and Anshul Kundaje for helpful discussions and sharing their code.

Conflict of Interest: None declared.

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
