# A  Appendix

## A.1  Illustration of group actions

This section is intended to provide a visual, more intuitive understanding of the different group actions on the tensors of our network. We begin with a visualization of the group action for the input space. We exemplify it over the sequence `GGACT`, whose reverse complement is `AGTCC`. The sequence is one hot encoded as explained in the main text and the group action over $\mathbb{Z}_2$ consist in flipping the tensor along the spatial axis and swapping the channels pairwise.

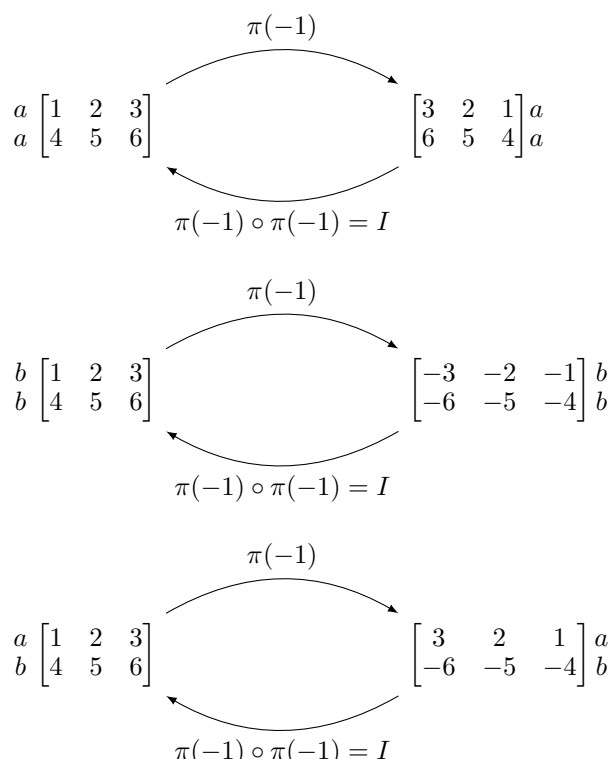

Now we illustrate the actions of other representations, on an example tensor $\begin{bmatrix} 1 & 2 & 3 \\ 4 & 5 & 6 \end{bmatrix}$ with two channels (of type $a$ or $b$) and three positions; this could typically be the representation of an input sequence of length 3 in an intermediate layer of dimention 2. Choosing the canonical representations of type $(I, 2, 0)$, $(I, 0, 2)$ and $(I, 1, 1)$ respectively, we get the following group actions (for clarity we add the channel type, $a$ or $b$, near each matrix row):

Finally, when using different values for P, we can get other group actions. As mentioned in the main text, by choosing $(P_{reg}, 1, 1)$, where $P_{reg} = \begin{bmatrix} 1 & 1 \\ 1 & -1 \end{bmatrix}$, we get the regular representation that flips the input channel. We also provide an example of the group action for a general P matrix,

by choosing $(P_{general}, 1, 1)$, where $P_{general} = \begin{bmatrix} 1 & 3 \\ 1 & -1 \end{bmatrix}$, we get a representation on the fibers $\rho_{general} = \begin{bmatrix} -0.5 & 1.5 \\ 0.5 & 0.5 \end{bmatrix}$

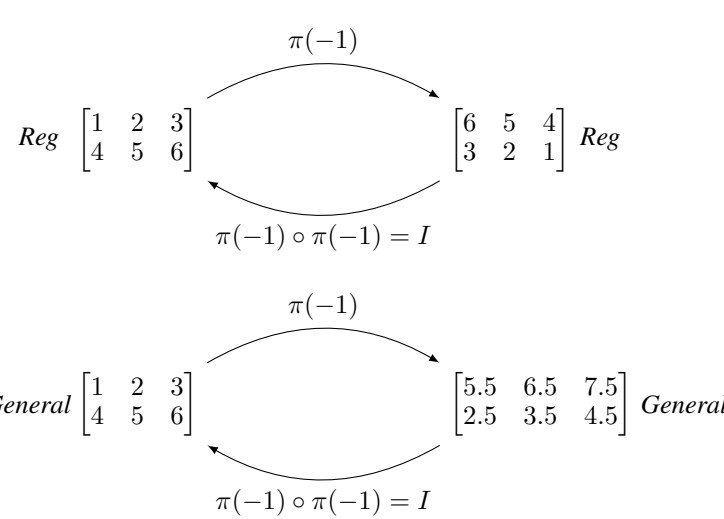

Over the course of these examples, we have limited ourselves to the case where the input tensor had only three nucleotides and two channels, but this is coincidental. The representation with arbitrary P can mix an arbitrary number of channels together with the group action.

## A.2 Proof of Theorem 1

*Proof.* The irreducible representations (irreps) of the 2-elements group $\mathbb{Z}_2$ are the 1-dimensional trivial and sign representations, given respectively by $\rho_1(s) = 1$ and $\rho_1(s) = s$. Any representation $\rho_n$ can be decomposed as a direct sum of irreps, and since each irrep is 1-dimensional this means that there exists an invertible matrix $P$ such that $P\rho_n(s)P^{-1}$ is diagonal, with diagonal terms either equal to 1 or equal to s. If we denote by $a_n$ (resp. $b_n$) the number of diagonal terms equal to 1 (resp. s), then Theorem 1 follows. □

## A.3 Proof of Theorem 2

*Proof.* Cohen et al. [11, Theorem 3.3] gives a general result about linear equivariant mapping. We first show that this result can be applied here, to show that these linear mappings are exactly the ones written as (2) and (3). For sake of clarity, we then provide a fully self-contained proof of the same result.

Let us first show that (2) and (3) correspond to a particular case of Cohen et al. [11, Theorem 3.3]. Under the notations of [11], our group is $G = \mathbb{Z} \rtimes \mathbb{Z}_2$, a locally compact, semi-direct product group. We choose $H = H_1 = H_2 = \mathbb{Z}_2$, making the coset space $G/H = \mathbb{Z}$. Since our group is a semi direct product group, we have $h_1(x, s) = s$. The spaces $F_n$ that we have considered are signals in $\mathbb{R}^D$ over the coset space, acted upon by the representation induced by $\rho$. Equivalently, they are sections of the associated vector bundle for the trivial case of a product group. Therefore, these $F_n$ exactly coincide with the setting of Cohen et al. [11, Theorem 3.3] and $\{\phi : F_n \to F_{n+1} | \pi_{n+1}\phi = \phi\pi_n\}$ is exactly $\mathcal{H}$. Then, by [11, Theorem 3.3], $\phi : F_n \to F_{n+1}$ is equivariant if and only if it can be written as a convolution:

$$\forall (f, x) \in F_n \times \mathbb{Z}, \quad \phi(f)(x) = \sum_{y \in \mathbb{Z}} \kappa(y - x) f(y), \tag{2}$$

where the kernel $\kappa : \mathbb{Z} \to \mathbb{R}^{D_{n+1} \times D_n}$ satisfies:

$$\forall x \in \mathbb{Z}\, s \in \mathbb{Z}_2,, \quad \kappa(sx) = \rho_{n+1}(s)\kappa(x)\rho_n(s^{-1}). \tag{6}$$

Using that for $s \in \mathbb{Z}_2$, $s^{-1} = s$, and the triviality of this equation for $s = 1$, we get that (6) is equivalent to (3)

For sake of clarity and completeness, we now provide a more explicit and self-contained proof for (2) and (3), that follows the one of [40, Theorem 2] in our specific setting. We first notice that any linear mapping $\phi; F_n \to F_{n+1}$ can be written as

$$\forall (f, x) \in F_n \times \mathbb{Z}, \quad \phi(f)(x) = \sum_{y \in \mathbb{Z}} k(x, y) f(y),$$

for some function $k : \mathbb{Z}^2 \to \mathbb{R}^{d_{n+1} \times d_n}$. For any $g = ts \in G$, the action of $G$ on $F_{n+1}$ gives:

$$\forall (f, x) \in F_n \times \mathbb{Z}, \quad \pi_{n+1}(g)\phi(f)(x) = \rho_{n+1}(s)\phi(f)(s(x - t))$$
$$= \rho_{n+1}(s) \sum_{y \in \mathbb{Z}} k(s(x - t), y) f(y). \tag{7}$$

Similarly, the action of $G$ on $F_n$ followed by $\phi$ gives:

$$\forall (f, x) \in F_n \times \mathbb{Z}, \quad \phi(\pi_n(g)f)(x) = \sum_{y \in \mathbb{Z}} k(x, y)\pi_n(g)f(y)$$
$$= \sum_{y \in \mathbb{Z}} k(x, y)\rho_n(s)f(s(y - t)) \tag{8}$$
$$= \sum_{y \in \mathbb{Z}} k(x, sy + t)\rho_n(s)f(y)$$

where we made the change of variable $y \mapsto sy + t$ to get the last equality. $\phi$ is equivariant if and only if, for any $g \in G$, $\phi \circ \pi_n(g) = \pi_{n+1}(g) \circ \phi$, which from (7) and (8) is equivalent to:

$$\forall (f, x) \in F_n \times \mathbb{Z}, \quad \rho_{n+1}(s) \sum_{y \in \mathbb{Z}} k(s(x - t), y) f(y) = \sum_{y \in \mathbb{Z}} k(x, sy + t)\rho_n(s)f(y). \tag{9}$$

For any $y_0 \in \mathbb{Z}$ and $v \in \mathbb{R}^{D_n}$, let us apply this equality to the function $f \in F_n$ given by $f(y_0) = v$ and $f(y) = 0$ for $y \neq y_0$:

$$\forall (x, y_0, v) \in \mathbb{Z} \times \mathbb{Z} \times \mathbb{R}^{D_n}, \quad \rho_{n+1}(s)k(s(x - t), y_0)v = k(x, sy_0 + t)\rho_n(s)v.$$

Since this must hold for any $v \in \mathbb{R}^{D_n}$ this necessarily implies:

$$\forall (x, y_0) \in \mathbb{Z}^2, \quad \rho_{n+1}(s)k(s(x - t), y_0) = k(x, sy_0 + t)\rho_n(s).$$

With the change of variable $y = s(y_0 - t)$, this is equivalent to:

$$\forall (x, y) \in \mathbb{Z}^2, \quad \rho_{n+1}(s)k(s(x - t), s(y - t)) = k(x, y)\rho_n(s),$$

which itself is equivalent to

$$\forall (x, y) \in \mathbb{Z}^2, \quad k(s(x - t), s(y - t)) = \rho_{n+1}(s)k(x, y)\rho_n(s), \tag{10}$$

where we used the fact that $\rho_{n+1}(s)^2 = \rho_{n+1}(s^2) = I$ for any $s \in \mathbb{Z}_2$. This must hold in particular for $s = 1$ and $t = x$, which gives:

$$\forall (x, y) \in \mathbb{Z}^2, \quad k(0, y - x) = k(x, y),$$

i.e., $k$ is necessarily translation invariant in the sense that there must exist a function $\kappa : \mathbb{Z} \to \mathbb{R}^{D_{n+1} \times D_n}$ such that

$$\forall (x, y) \in \mathbb{Z}^2, \quad k(x, y) = \kappa(y - x).$$

From (10) we see that $\kappa$ must satisfy

$$\forall (x, y) \in \mathbb{Z}^2, \quad \kappa(s(y - x)) = \rho_{n+1}(s)\kappa(y - x)\rho_n(s),$$

which boils down to the following constraint, after observing that the constraint is always true for $s = 1$ and is therefore only nontrivial for $s = -1$:

$$\forall x \in \mathbb{Z}, \quad \kappa(-x) = \rho_{n+1}(-1)\kappa(x)\rho_n(-1). \tag{11}$$

At this point, we have therefore shown that an equivariant linear function must have an expansion of the form

$$\forall (f, x) \in F_n \times \mathbb{Z}, \quad \phi(f)(x) = \sum_{y \in \mathbb{Z}} \kappa(y - x) f(y),$$

where $\kappa$ must satisfy (11). Conversely, such a linear layer trivially satisfies (9), and is therefore equivariant. This proves (2) and (3).

To prove (4), we simply rewrite (3) using Theorem 1:

$$\forall x \in \mathbb{Z}, \quad \kappa(-x) = P_{n+1}\mathrm{Diag}(I_{a_{n+1}}, -I_{b_{n+1}})P_{n+1}^{-1}\kappa(x)P_n\mathrm{Diag}(I_{a_n}, -I_{b_n})P_n^{-1}. \tag{12}$$

Thus writing the matrix $K = P_{n+1}^{-1}\kappa(x)P_n$ by blocs of sizes $a_{n+1} \times a_n$, $a_{n+1} \times b_n$, $b_{n+1} \times a_n$ and $b_{n+1} \times b_n$, we have :

$$\begin{aligned}
(12) &\iff K(-x) = \mathrm{Diag}(I_{a_{n+1}}, -I_{b_{n+1}})K(x)\mathrm{Diag}(I_{a_n}, -I_{b_n}) \\
&\iff \begin{bmatrix} \alpha(-x) & \beta(-x) \\ \gamma(-x) & \delta(-x) \end{bmatrix} = \begin{bmatrix} \alpha(x) & -\beta(x) \\ -\gamma(x) & \delta(x) \end{bmatrix}
\end{aligned}$$

This gives us the equivalence $(3) \iff (12) \iff (4)$. $\qquad\square$

## A.4 Resolution of the constraint for other basis

To go from an arbitrary representation $(P, a, b)$ to another, we can write an odd/even kernel and change of basis. One may also solve the constraints (3) for specific representations, and save the need of multiplication by $P_{n+1}$ and $P_n^{-1}$ in (4). In this section, we solve the constraint in other basis, to go from one kind of representation (irrep or regular) to another. We just substitute the correct representation and see what constrained kernel it gives. The irrep and regular representations are in a basis such that they write as :

$$\rho_{irrep} = \begin{bmatrix} I_a & 0 \\ 0 & -I_b \end{bmatrix}, \quad \rho_{reg} = \begin{bmatrix} 0 & 0 & \dots & 1 \\ \vdots & & & \vdots \\ 0 & 1 & \dots & 0 \\ 1 & 0 & \dots & 0 \end{bmatrix}.$$

We get the following table of constraints :

| $F_n$ \ $F_{n+1}$ | 'irrep' | 'regular' |
|---|---|---|
| 'irrep' | $\begin{bmatrix} \alpha(-x) & \beta(-x) \\ \gamma(-x) & \delta(-x) \end{bmatrix} = \begin{bmatrix} \alpha(x) & -\beta(x) \\ -\gamma(x) & \delta(x) \end{bmatrix}$ | $[\kappa_{j,a}(-x), \kappa_{j,b}(-x)] = [\kappa_{n-j,a}(x), -\kappa_{n-j,b}(x)]$ |
| 'regular' | $\begin{bmatrix} \kappa_{a,j}(-x) \\ \kappa_{b,j}(-x) \end{bmatrix} = \begin{bmatrix} \kappa_{a,n-j}(x) \\ -\kappa_{b,n-j}(x) \end{bmatrix}$ | $\kappa_{i,j}(-x) = -\kappa_{n-i,n-j}(x)$ [34] |

## A.5 Proof of Theorem 3

With a slight abuse of notations, in this section we denote the matrix $\rho(-1)$ simply by $\rho \in \mathbb{R}^{D \times D}$, and for any $\theta : \mathbb{R} \to \mathbb{R}$ we define $\tilde{\theta}(x) := \theta(x) - \theta(0)$. We start with three technical lemmas, before proving Theorem 3.

**Lemma 4.** *Let $h : \mathbb{R} \to \mathbb{R}$ be a continuous function with left and right derivatives at 0. If there exists $A \in \mathbb{R}$ with $|A| > 1$ such that*

$$\forall x \in \mathbb{R}, \quad h(x) = Ah(A^{-1}x), \tag{13}$$

*then $h$ is a leaky ReLu function, i.e., there exists $(\alpha_-, \alpha_+) \in \mathbb{R}^2$ such that*

$$\forall x \in \mathbb{R}, \quad h(x) = \begin{cases} \alpha_- x & \text{if } x \leq 0, \\ \alpha_+ x & \text{if } x \geq 0. \end{cases}$$

*In addition, if $A < -1$, then $\alpha_- = \alpha_+$, i.e., $h$ is linear.*

*Proof.* Equation (13) implies $h(0) = 0$ and

$$\forall x \in \mathbb{R}^*, \quad \frac{h(x)}{x} = \frac{h(A^{-1}x)}{A^{-1}x},$$

which by simple induction gives more generally:

$$\forall (x, n) \in \mathbb{R}^* \times \mathbb{N}, \quad \frac{h(x)}{x} = \frac{h(A^{-n}x)}{A^{-n}x}. \tag{14}$$

The right-hand side of (14) for $n = 2k$ converges to $h'_{sign(x)}(0)$ when $k \to +\infty$, which by unicity of the limit must be equal to the left-hand side. As a result, for any $x \in \mathbb{R}$, $h(x) = h'_{sign(x)}(0)x$, i.e., $h$ is a leaky ReLu function with $\alpha_s = h'_s(0)$ for $s \in \{-, +\}$. If in addition $A < -1$, then (14) for $n = 2k + 1$ converges to $h'_{-sign(x)}(0)$ when $k \to +\infty$. By unicity of the limit, this implies $h'_-(0) = h'_+(0)$, i.e., $\alpha_- = \alpha_+$. $\qquad \square$

**Lemma 5.** *Under the assumptions of Theorem 3, if $\bar{\theta}_F$ is equivariant and if there exists $(i, j) \in [1, D]^2$ such that $\rho_{ij} \notin \{-1, 0, 1\}$, then necessarily $\tilde{\theta}$ is a leaky ReLu function.*

*Proof.* For any $(i, j)$, applying the equivariance constraint $\theta(\rho x)_i = \rho\theta(x)_i$ to the vector $x = ae_j$, for any $a \in \mathbb{R}$, gives the equation:

$$\forall a \in \mathbb{R}, \quad \theta(a\rho_{ij}) = \rho_{ij}\theta(a) + (\sum_{k \neq j} \rho_{ik})\theta(0).$$

If $|\rho_{ij}| > 1$, we can rewrite it as

$$\forall a \in \mathbb{R}, \quad \theta(a) = \rho_{ij}\theta(a\rho_{ij}^{-1}) + (\sum_{k \neq j} \rho_{ik})\theta(0),$$

and if $0 < |\rho_{ij}| < 1$ we can rewrite it as

$$\forall a \in \mathbb{R}, \quad \theta(a) = \rho_{ij}^{-1}\theta(a\rho_{ij}) - \rho_{ij}^{-1}(\sum_{k \neq j} \rho_{ik})\theta(0).$$

In both cases, this is an equation of the form

$$\forall a \in \mathbb{R}, \quad \theta(a) = A\theta(A^{-1}a) + B,$$

where $|A| > 1$. Subtracting to this equation the same equation written for $a = 0$ gives

$$\forall a \in \mathbb{R}, \quad \tilde{\theta}(a) = A\tilde{\theta}(A^{-1}a). \tag{15}$$

By Lemma 4, $\tilde{\theta}$ is a leaky ReLu function.

$\qquad \square$

**Lemma 6.** *Under the assumptions of Theorem 3, if $\bar{\theta}_F$ is equivariant and if there exists at least one row in $\rho$ with at least two nonzero entry, then necessarily $\theta$ is an affine function.*

*Proof.* Let us suppose that $\rho$ contains at least a row $i$ with two nonzero entries, say $\rho_{ij} \neq 0$ and $\rho_{ik} \neq 0$. Then taking $x = x_j e_j + x_k e_k$ with $x_j, x_k \in \mathbb{R}$, the equivariance constraint for the $i$-th dimension gives

$$\forall x_j, x_k \in \mathbb{R}, \quad \theta(\rho_{ij}x_j + \rho_{ik}x_k) = \rho_{ij}\theta(x_j) + \rho_{ik}\theta(x_k) + C\theta(0),$$

with $C = \sum_{p \notin \{j,k\}} \rho_{ip}$. Subtracting to this equation the same equation written for $x_j = x_k = 0$ allows us to remove the constant term and get

$$\forall x_j, x_k \in \mathbb{R}, \quad \tilde{\theta}(\rho_{ij}x_j + \rho_{ik}x_k) = \rho_{ij}\tilde{\theta}(x_j) + \rho_{ik}\tilde{\theta}(x_k). \tag{16}$$

We now prove that $\tilde{\theta}$ is necessarily a leaky ReLu function, i.e., that there exist $(\alpha_+, \alpha_-) \in \mathbb{R}^2$ such that $\tilde{\theta}(x) = \alpha_{sign(x)}x$, with potentially $\alpha_+ \neq \alpha_-$. By Lemma 5 this is true if $|\rho_{ij}| \neq 1$ or $|\rho_{ik}| \neq 1$, so we focus on the case $|\rho_{ij}| = |\rho_{ik}| = 1$, which we decompose in two subcases. First, if $\rho_{ij} = \rho_{ik} = s$ with $s \in \{-1, 1\}$, then taking $x_j = x_k = a$ in (16) gives $\tilde{\theta}(2sa) = 2s\tilde{\theta}(a)$, for any $a \in \mathbb{R}$. Second, if $\rho_{ij} = -\rho_{ik} = 1$ (resp. $\rho_{ij} = -\rho_{ik} = 1$), then taking $x_j = 2a$ and $x_k = a$ (resp.

$x_j = a$ and $x_k = 2a$) gives $\tilde{\theta}(2a) = 2\tilde{\theta}(a)$. In both subcases, by Lemma 4, $\tilde{\theta}$ must be a leaky ReLu function.

Knowing that $\tilde{\theta}$ is a leaky ReLu function with coefficients $\alpha_+$ and $\alpha_-$, in order to prove that $\theta$ is necessarily an affine function (i.e., that $\tilde{\theta}$ is linear), we need to show that $\alpha_+ = \alpha_-$. For that purpose, let us first suppose that $\rho_{ij}$ and $\rho_{ik}$ are both positive or both negative. Then there exists a pair $(x_j, x_k) \in \mathbb{R}^2$ such that $x_j > 0$, $x_k < 0$ and $\rho_{ij}x_j + \rho_{ik}x_k < 0$. Similarly, if $\rho_{ij}$ and $\rho_{ik}$ are of different signs, say without loss of generality $\rho_{ij} < 0$ and $\rho_{ik} > 0$, then any pair $(x_j, x_k) \in \mathbb{R}^2$ such that $x_j > 0$, $x_k < 0$ satisfies $\rho_{ij}x_j + \rho_{ik}x_k < 0$. In both cases, using the fact that $\tilde{\theta}$ is linear on $\mathbb{R}_+$ and on $\mathbb{R}_-$, (16) gives

$$\alpha_-(\rho_{ij}x_j + \rho_{ik}x_k) = \alpha_+\rho_{ij}x_j + \alpha_-\rho_{ik}x_k \, ,$$
$$\Longleftrightarrow \alpha_-\rho_{ij}x_j = \alpha_+\rho_{ij}x_j$$
$$\Longleftrightarrow \alpha_- = \alpha_+ \, .$$

$\square$

We are now ready to prove Theorem 3.

***Proof of Theorem 3.*** To characterize the functions $\theta$ and representations $\rho$ such that $\bar{\theta}_F$ is equivariant, we proceed by a disjunction of cases on $\theta$, depending on whether it is affine.

If $\theta$ is affine, say $\theta(x) = \alpha x + \beta$, then $\bar{\theta}_F$ is equivariant if and only if, for any $x \in \mathbb{R}^D$, $\bar{\theta}_{\mathbb{R}^D}(\rho x) = \rho\bar{\theta}_{\mathbb{R}^D}(x)$. This is equivalent to

$$\forall (i, x) \in [1, d] \times \mathbb{R}^D, \quad \sum_{j=1}^{D} \rho_{i,j}\theta(x_j) = \theta\left(\sum_{j=1}^{D} \rho_{i,j}x_j\right)$$

$$\Longleftrightarrow \forall (i, x) \in [1, d] \times \mathbb{R}^D, \quad \sum_{j=1}^{D} \rho_{i,j}(\alpha x_j + \beta) = \alpha\left(\sum_{j=1}^{D} \rho_{i,j}x_j\right) + \beta$$

$$\Longleftrightarrow \forall i \in [1, d], \quad \beta\left(\sum_{j=1}^{D} \rho_{i,j} - 1\right) = 0 \, .$$

This shows that if $\theta$ is affine, then $\bar{\theta}_F$ is equivariant if and only $\beta = 0$, i.e., $\theta$ is linear (case 1 of Theorem 3), or $\rho\mathbf{1} = \mathbf{1}$ (case 2 of Theorem 3).

If $\theta$ is not affine and $\bar{\theta}_F$ is equivariant, then by Lemma 6 we know that $\rho$ can have at most one nonzero entry per row. Since $\rho$ is invertible, it must have at least one nonzero entry per row, so we conclude that if contains exactly one nonzero entry per row, hence a total of $D$ nonzero entries. Being invertible, it must also contain at least one nonzero entry per column, so we conclude that it contains also exactly one nonzero entry per column. Using the fact that $\rho^2 = I$, we can further clarify how nonzero entries must be organized:

- For a nonzero entry $\rho_{ii} \neq 0$ on the diagonal, we must have $\rho_{ii}^2 = 1$, i.e., $\rho_{ii} \in \{-1, +1\}$.

- For an off-diagonal nonzero entry $\rho_{ij} \neq 0$ with $i \neq j$, we must have $\rho_{ij}\rho_{ji} = 1$, i.e., $\rho_{ji} = \rho_{ij}^{-1}$.

Splitting the nonzero entries by sign, this implies that there exists a permutation matrix $\Pi$ such that

$$\hat{\rho} := \Pi^{-1}\rho(-1)\Pi = \bigoplus_{i=1}^{a} \begin{pmatrix} 0 & \lambda_i \\ \lambda_i^{-1} & 0 \end{pmatrix} \oplus \bigoplus_{i=1}^{b} \begin{pmatrix} 0 & -\mu_j \\ -\mu_j^{-1} & 0 \end{pmatrix} \oplus (1)^{\oplus c} \oplus (-1)^{\oplus d}, \qquad (17)$$

for some $(a, b, c, d) \in \mathbb{N}^4$ such that $a + b + c + d = D$ and $(\lambda, \mu) \in \mathbb{R}_+^a \times \mathbb{R}_+^b$. For any $i \in [1, D]$, let us now denote by $\tau(i)$ the column corresponding to the nonzero entry of the $i$-th row of $\hat{\rho}$, i.e.,

the only index such that $\hat{\rho}_{i\tau(i)} \neq 0$. Then the action of $\hat{\rho}$ on a vector $v \in \mathbb{R}^D$ has the simple form $[\hat{\rho}v]_i = \hat{\rho}_{i\tau(i)}v_{\tau(i)}$. By writing the equivariance property $\rho \circ \bar{\theta}_F = \bar{\theta}_F \circ \rho$ coordinate by coordinate, we can therefore say that $\bar{\theta}_F$ is equivariant if and only if:

$$\forall (i,x) \in [1,D] \times \mathbb{R}, \quad \theta(\hat{\rho}_{i\tau(i)}x) = \hat{\rho}_{i\tau(i)}\theta(x). \tag{18}$$

Let us now consider two possible cases:

- If there exists $i \in [1,D]$ such that $|\hat{\rho}_{i\tau(i)}| \neq 1$, then by Lemma 5 $\tilde{\theta}$ is a leaky ReLu function, i.e., there exist $(\alpha_+, \alpha_-, \beta) \in \mathbb{R}^3$ such that $\forall x \in \mathbb{R}, \theta(x) = \alpha_{sign(x)}x + \beta$. In that case, by (18), $\bar{\theta}_F$ is equivariant if and only if:

$$\forall (i,x) \in [1,D] \times \mathbb{R}, \quad \alpha_{sign(\hat{\rho}_{i\tau(i)}x)}\hat{\rho}_{i\tau(i)}x + \beta = \hat{\rho}_{i\tau(i)}\left(\alpha_{sign(x)}x + \beta\right),$$

$$\iff \forall i \in [1,D], \quad \begin{cases} \alpha_{sign(\hat{\rho}_{i\tau(i)})} = \alpha_+, \\ \alpha_{sign(-\hat{\rho}_{i\tau(i)})} = \alpha_-, \\ \beta = \hat{\rho}_{i\tau(i)}\beta, \end{cases} \tag{19}$$

$$\iff \begin{cases} \forall i \in [1,D], \quad \alpha_{sign(\hat{\rho}_{i\tau(i)})} = \alpha_+, \\ \beta = 0, \end{cases}$$

where the first equivalence comes from identifying the coefficients of the linear equation in $x$ on $\mathbb{R}_-$ and $\mathbb{R}_+$, and the second equivalence comes from the observation that the two conditions in $\alpha$ in the first equivalence are themselves equivalent to each other, so we can keep only one of them, and that the condition on $\beta$ is equivalent to $\beta = 0$ since we assume the existence of an $i \in [1,D]$ such that $\hat{\rho}_{i\tau(i)} \neq 1$. Since we assume that $\theta$ is not affine, we can not have $\alpha_- = \alpha_+$, which by (19) rules out the possibility of having negative entries in $\hat{\rho}$, i.e., necessarily $b = d = 0$ in (17). If that is not the case, then the condition on $\alpha$ in (19) is automatically met for all $i \in [1,D]$, so we have that $\bar{\theta}_F$ is equivariant if and only if $\beta = 0$, i.e., if and only if $\theta$ is a leaky ReLu function. This is the second statement in Case 3 of Theorem 3, when we further notice that when $b = 0$ the only entry in $\hat{\rho}$ that can have been different from -1 and 1 is a $\lambda_i$ in (17).

- If for all $i \in [1,D], |\hat{\rho}_{i\tau(i)}| = 1$, then (17) simplifies as

$$\hat{\rho} = \bigoplus_{i=1}^{a}\begin{pmatrix} 0 & 1 \\ 1 & 0 \end{pmatrix} \oplus \bigoplus_{i=1}^{b}\begin{pmatrix} 0 & -1 \\ -1 & 0 \end{pmatrix} \oplus (1)^{\oplus c} \oplus (-1)^{\oplus d}.$$

In that case, the equivariance condition (18) is particularly simple, and true for any $\theta$ for positive values. For each $i$ such that $\hat{\rho}_{i\tau(i)} = -1$ it reads $\forall x \in \mathbb{R}, -\theta(x) = \theta(-x)$, and is therefore true if and only if $\theta$ is odd. Noticing that the latter constraint occurs if and only if $b + d > 0$ finally leads to the first and third statements in Case 3 of Theorem 3.

$\square$

## A.6 Additional result

### A.6.1 Effect of data augmentation and size for non-equivariant models

Given a non-equivariant model, a simple way to let it "learn" to be equivariant is to train it with data augmentation, where for each sequence in the training set we add its reverse complement to the training set. This doubles the size of the training set, which increases the training time. If we compare such a non-equivariant model with an equivariant model with the same number of channels in each layers, then it has about twice the same number of free parameters to train, and we therefore call it "big"; as an alternative, one may want to restrict the number of channels in each layer to enforce the same number of parameters as the equivariant model. To assess the benefits of data augmentation and number of channels, we plot in Figure 6 the performance of a standard, non-equivariant model with or without data augmentation, and with the same number of channels or half of it, on the binary classification tasks. We see that the number of channels has no significant impact on the performance, but that data augmentation has a significant positive impact. In the main text, we therefore restrict ourselves to the standard model with data augmentation as non-equivariant baseline model.

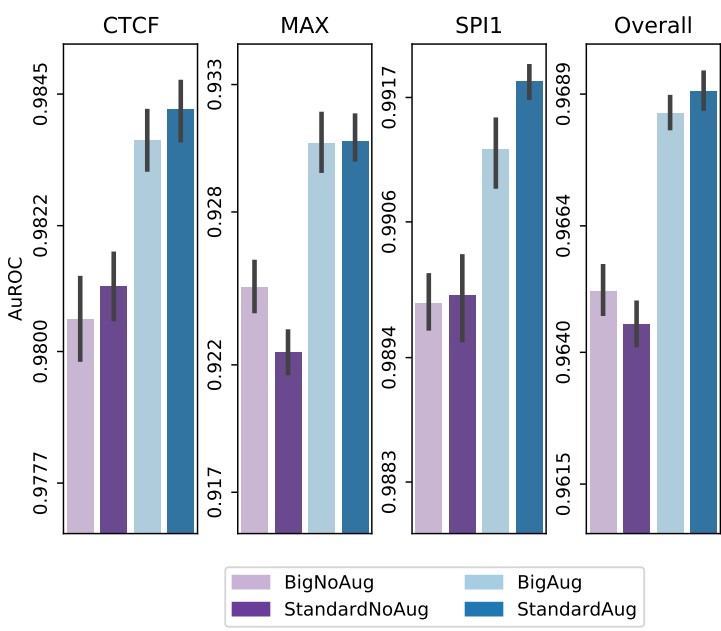

Figure 6: Binary task performance of a standard, non-equivariant model trained with ("Aug") or without ("NoAug") data augmentation, and with more ("Big") or less ("Standard") channels.

### A.6.2 Comparison of learning curves

Because equivariant model are supposed to converge faster, we looked into the learning curves of our models, i.e., how the test performance increases as a function of the number of epochs during training. However, we do not see a major difference in the learning dynamics between the equivariant and non equivariant models.

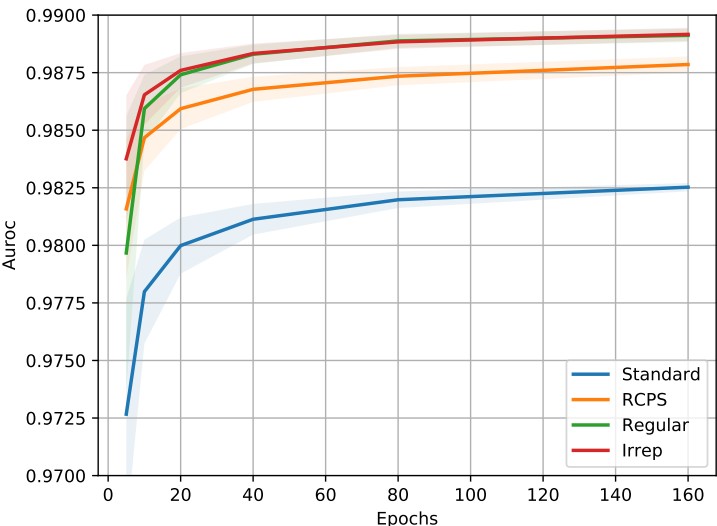

Figure 7: AuROC performance of the four different models on the three binary classification problems CTCF, MAX and SPI1, as well as their average over the course of learning.