# OpenReview forum: "Reverse-Complement Equivariant Networks for DNA Sequences"
_NeurIPS.cc/2021/Conference — NeurIPS 2021 Poster_

### Official Review · Reviewer_YzSz · 2021-07-15

**Rating:** 6
**Confidence:** 4

**Summary:**

The authors note that neural network models operating on DNA sequences desire shift equivariance and reverse complement (RC) equivariance. Existing proposed techniques have been somewhat ad hoc. The authors derive RC equivariant models from first principles, and expose a class of techniques that haven’t yet been studied. They benchmark several equivariant alternatives on ChIP-seq peak classification and nucleotide-resolution coverage prediction and demonstrate interesting results.

**Limitations And Societal Impact:**

The authors do not describe potential negative societal impacts, but I cannot imagine any.

**Main Review:**

Introductory note
---------

I’m unable to evaluate the correctness of the Methods section due to unfamiliarity with group theory.

Major comments
---------

In describing their experiments, the authors note, “For a more detailed explanation of the experimental setup, please refer to Zhou et al. [44].” For some of the details (e.g. the ChIP-seq data processing pipeline), I think that’s fine. But there are several important choices that I believe need to be described here. First, it is standard when performing deep learning on these data to include both the sequence and its reverse complement in the training set, akin to data augmentation. If this has not been done, then the “Standard” model would be at an unfair disadvantage.

Similarly, it is typical when making predictions to compute a forward pass for both the forward and reverse complement sequences and take the average as an ensemble. An equivariant model wouldn’t benefit from this option. However, you might argue that it would be fair to benchmark an ensemble of two distinctly trained equivariant models to match the prediction computation to that mode. The authors should offer more details about how predictions are made on their test sets, and I’d encourage them to explore these different modes.

Minor comments
---------
The authors might have more success reaching potential users of this technique in computational biology if they could make their Methods section more accessible to non-experts in group theory. If the authors could convey some intuition for what these operations aim to accomplish and how they do it, I believe a larger audience would be available. I’ll note that the code availability is great and a dedicated reader can likely reach an understanding.

In line 107, the authors write that “the RC operations amounts to the following : RC(f)(x) = σ(−1)[f(−x)], where σ(−1) is the 4×4 permutation matrix that exchanges complementary bases A/T and C/G”. Is the negative sign in front of x to be explicitly interpreted as changing the one hot encoded vectors like [0,1,0,0] to [0,-1,0,0]. I expected some mathematical treatment of reversing the sequence, but I don’t understand how this negating would accomplish that.

NeurIPS dimensions
---------

Originality: The method here represents a novel derivation of equivariant reverse complement DNA deep learning layers. It generalizes prior formulations and suggests alternatives, which are benchmarked.

Quality: The submission is technically sound, and the claims are supported by experiments, although these experiments need to be clarified according to my comments above. The authors do not carefully and honestly evaluate the weaknesses of their work.

Clarity: The submission is challenging to follow for readers without group theory expertise. For those who have such expertise, it seems to be reasonably well-organized.

Significance: Deep learning on DNA sequences has many important applications, and improvements to how it’s done can be quite significant. Exploring the reverse complement equivariance with this mathematical approach is highly informative, but the results suggest that the value is not tremendous and large data applications may choose to forgo this complicated technique.

Update after rebuttal
---------
The authors have greatly clarified my points of concern, and their experiments now clearly demonstrate interesting performance versus alternatives.

**Time Spent Reviewing:**

5

---

> ### Author Response · Authors · 2021-08-10
> **Answer to Reviewer #3**
>
> We thank Reviewer #3 for his review. We refer to the general answer to reviewers for a more general answer to the reviews. However here are the point by point answers to Reviewer #3 ‘s questions.
>
> > In describing their experiments, the authors note, “For a more detailed explanation of the experimental setup, please refer to Zhou et al. [44].” For some of the details (e.g. the ChIP-seq data processing pipeline), I think that’s fine. But there are several important choices that I believe need to be described here. First, it is standard when performing deep learning on these data to include both the sequence and its reverse complement in the training set, akin to data augmentation. If this has not been done, then the “Standard” model would be at an unfair disadvantage.
>
> Zhou et al. [44] have shown that, indeed, reverse complement data augmentation improves the performance of the “Standard” model (at the cost of more computation); however, it is significantly worse than RCPS in performance, even after data augmentation. Since we use the same benchmark, we did not feel the need to include the result to experiment, but following the reviewer’s remark we agree that this is a result we could add as supplementary figure so that the reader has a clear view of a stronger baseline (we re-ran the “Standard” model with data augmentation with our code and obtained the same results as Zhou et al [44]).
>
> > Similarly, it is typical when making predictions to compute a forward pass for both the forward and reverse complement sequences and take the average as an ensemble. An equivariant model wouldn’t benefit from this option. However, you might argue that it would be fair to benchmark an ensemble of two distinctly trained equivariant models to match the prediction computation to that mode. The authors should offer more details about how predictions are made on their test sets, and I’d encourage them to explore these different modes.
>
> As noticed by the reviewer, the equivariant architectures we compare would not benefit from aggregating predictions on a forward and
> backward strand, since the predictions would be the same by design. We agree that performance is likely to increase if we train several networks with different initializations and aggregate their predictions. However, since this is a general phenomenon of deep models not related to the presence of symmetries in the data, we focused our experiments on comparing models without ensemble in order to focus on the benefits of taking into account symmetries in the data. We agree that it would be an interesting future work to investigate how aggregation boosts the performance of different tasks, for both equivariant and non-equivariant models.
>
> > The authors might have more success reaching potential users of this technique in computational biology if they could make their Methods section more accessible to non-experts in group theory. If the authors could convey some intuition for what these operations aim to accomplish and how they do it, I believe a larger audience would be available. I’ll note that the code availability is great and a dedicated reader can likely reach an understanding.
>
> Indeed we decided to provide a rigorous mathematical exposition of our work, and understand that many readers may not be familiar with the math involved. We agree that adding a figure to give an intuitive understanding of the different equivariant layers we describe could be useful, and will do so in the final version. To help users with the code, we also plan to add more examples in the Github repository.
>
> > In line 107, the authors write that “the RC operations amounts to the following : RC(f)(x) = σ(−1)[f(−x)], where σ(−1) is the 4×4 permutation matrix that exchanges complementary bases A/T and C/G”. Is the negative sign in front of x to be explicitly interpreted as changing the one hot encoded vectors like [0,1,0,0] to [0,-1,0,0] ? I expected some mathematical treatment of reversing the sequence, but I don’t understand how this negating would accomplish that.
>
> In our notations, $x$ represents the spatial position (an integer) and $f(x)$ represents the signal (a vector) at position $x$, e.g., $f(10)=[0,1,0,0]$. So the minus sign in front of $x$ denotes a change in position, e.g, $\text{RC}(f)(50) = \sigma(-1)[f(-50)]$, and $\sigma(-1)$ is the matrix that changes the coordinates of $f(-x)$.
>
> However, we thank the reviewer for pointing out a little imprecision : in this mathematical framework, the applied convolution yields a signal that is not in the same spatial domain (either on the negative or positive values).
> To resolve this in a rigorous way, we need to switch the signal functions $f$ for equivalence classes of these signals. We represent sequences using equivalence classes of compact support functions that align their leftmost non zero values. This imprecision will be corrected.
>
> We thank again Reviewer #3.
>
> Best regards,
> The authors

---

> > ### Comment · Reviewer_YzSz · 2021-08-17
> > **RCPS versus data augmentation**
> >
> > Thanks to the authors for their in depth rebuttal to our reviewer comments. I very much look forward to the addition of a "figure to give an intuitive understanding of the different equivariant layers we describe". (Although as primarily a bioinformatic journal reviewer, I find it bizarre that I'm expected to update my review based merely on that sentence rather than seeing the figure.)
> >
> > I want to first clarify that my first major comment requests clarification in the text about how you conducted your experiments, as opposed to simply referencing Zhou et al. Benchmarking against alternative strategies would be nice, but the most important addition you can make is more description of your experiments.
> >
> > I'm also confused by your comment that, "Zhou et al. [44] have shown that, indeed, reverse complement data augmentation improves the performance of the “Standard” model (at the cost of more computation); however, it is significantly worse than RCPS in performance, even after data augmentation."
> >
> > From the abstract of Zhou et al, they appear to be stating the opposite, "Here we extend conjoined & RCPS models to base-resolution signal prediction, and introduce a strong baseline: a standard model (trained with RC data augmentation) that is made conjoined only after training, which we call “post-hoc” conjoined. Through benchmarks on diverse tasks, we find post-hoc conjoined consistently performs best or second-best, surpassed only occasionally by RCPS, and never underperforms conjoined-during-training. We propose an overfitting-based hypothesis for the latter finding, and study it empirically. Despite its theoretical appeal, RCPS shows mediocre performance on several tasks, even though (as we prove) it can represent any solution learned by conjoined models. Our results suggest users interested in RC equivariance should default to post-hoc conjoined as a reliable baseline before exploring RCPS"

---

> > > ### Author Response · Authors · 2021-08-23
> > > **Answer to Reviewer #3**
> > >
> > > We thank the reviewer for the new comments, and wish to provide a few clarifications below.
> > >
> > > > I want to first clarify that my first major comment requests clarification in the text about how you conducted your experiments, as opposed to simply referencing Zhou et al. Benchmarking against alternative strategies would be nice, but the most important addition you can make is more description of your experiments.
> > >
> > > Thank you for this clarification. We agree that simply referring to Zhou et al. [44] about the experimental protocol may not be sufficient to help the reader quickly understand what we did, and will update the corresponding paragraph (l.270-271) to be more precise on what we did:
> > >
> > > “ *We train all equivariant models without data augmentation. As a simple baseline to illustrate the benefit of equivariance, we add in the comparison a standard non-equivariant network with the same number of parameters as the RCPS model (i.e., with 16 filters while equivariant architectures have 32 filters, whose parameters are tied together by the equivariant constraints), also trained without data augmentation. For sake of completeness, we also include in Supplementary Section XX a more complete set of results with alternative data augmentation and pooling strategies investigated by Zhou et al. [44].* “
> > >
> > > For sake of completeness, we will also add a new Supplementary Section to recall the results obtained by Zhou et al. [44] (summarized in Figure 4 of https://www.biorxiv.org/content/10.1101/2020.11.04.368803v2.full.pdf ), with explanations to elucidate any doubts the reader might have about the training procedures :
> > >
> > > “*In this section we reproduce the major results obtained by Zhou et al. [44] using various strategies to handle the RC symmetry (including data augmentation and post-hoc pooling) and compare them to the equivariant models studied in this paper. In particular we include the performance of a standard non-equivariant model (with 16 filters) trained with and without data augmentation (referred to Standard-NoRCAug and Standard-RCAug in [44]), a standard model without data augmentation but with 32 filters instead of 16, in order to have the same number of filters (as opposed to the same number of parameters) as the RCPS model, and the “post-hoc conjoined” model of [44] trained with data augmentation and which uses post-hoc bagging of the prediction on a sequence and its reverse complement at test time.* ”
> > >
> > > To anticipate the new Figure, we provide below the AuROC performance over the binary task :
> > >
> > > | Method :   | Result on CTCF | Result on MAX | Result on SPI1  | Overall Result |
> > > |---|---:|---:|---:|---:|
> > > | Standard No Data Augmentation | 0.9827  | 0.9826 | 0.9824  | 0.9825 |
> > > | Standard Data Augmentation  | 0.9876 | 0.9881 | 0.9866 | 0.9874 |
> > > | More filters Data Augmentation | 0.9869 | 0.9855 | 0.9855 | 0.9863 |
> > > | RCPS |  0.9881 | 0.9866  | 0.9888 | 0.9879 |
> > > | Irrep | 0.9902 | 0.9891 |  0.9882 | 0.9892 |
> > > | Standard Data Augmentation  + Post-hoc pooling | 0.9904 | 0.9906 | 0.9901 | 0.9904 |
> > >
> > > “*As reported by Zhou et al. [44], we notice that adding data augmentation at training time for a non-equivariant model increases the performance, which makes it on average very close to the RCPS equivariant network but significantly worse than the newly proposed Irrep one. Adding more filters to this standard network seems to have a negative effect on performance. Like Zhou et al. [44], we find that the model with data augmentation at train time and post-hoc pooling at test time is very competitive and performs best. It should be noted, however, that using data augmentation doubles the training time, and that post-hoc bagging doubles the inference time. At a fixed level of computational cost, the Irrep equivariant model therefore performs best on this benchmark. Trading computational efficiency for increased performance through bagging is a well-known phenomenon; a more detailed investigation of the bagging effect on the performance of equivariant and non-equivariant models, if one is ready to double the computation time at prediction, is an interesting research direction for the future.*
> > > “
> > >
> > > We hope these explanations clarify the details of the experimental protocol, and would be happy to answer any remaining question.
> > >
> > > > I'm also confused by your comment that, "Zhou et al. [44] have shown that, indeed, reverse complement data augmentation improves the performance of the “Standard” model (at the cost of more computation); however, it is significantly worse than RCPS in performance, even after data augmentation."
> > >
> > > > From the abstract of Zhou et al, they appear to be stating the opposite, "Here we extend conjoined & RCPS models to base-resolution signal prediction, and introduce a strong baseline: a standard model (trained with RC data augmentation) that is made conjoined only after training, which we call “post-hoc” conjoined. Through benchmarks on diverse tasks, we find post-hoc conjoined consistently performs best or second-best, surpassed only occasionally by RCPS, and never underperforms conjoined-during-training. We propose an overfitting-based hypothesis for the latter finding, and study it empirically. Despite its theoretical appeal, RCPS shows mediocre performance on several tasks, even though (as we prove) it can represent any solution learned by conjoined models. Our results suggest users interested in RC equivariance should default to post-hoc conjoined as a reliable baseline before exploring RCPS"
> > >
> > > Thanks for this interesting comment. We would like to clarify that there is no contradiction because the “strong baseline” that Zhou et al. mention in the abstract combines two different components, both of which contribute to the good performance. They mention: 1) the use of data augmentation at train time, and 2) the use of model averaging (a.k.a. “pooling” or “bagging”) at test time.
> > >
> > > Our comment refers to the benefits of data augmentation at train time, when a standard (non-equivariant) network is trained; in other words, for each sequence/label pair in the training set, we add to the training set the reverse complement of the sequence with the same label (or with a reversed profile for the profile prediction task). This is indeed the standard way to “help” a model learn that it should predict the same output given a sequence or its reverse complement. This adds computation at training time (since the training set doubles in size). In Figure 4 of [44], RCPS significantly outperforms a standard model trained with data augmentation (referred to as "Standard-RCAug" in [44]) in all binary tasks except MAX, and incidentally we show that RCPS is indeed quite poor compared to other equivariant architectures we propose. Thus despite more computations, data augmentation performs worse than RCPS, with a significant difference.
> > >
> > > The second component is to combine the predictions, at test time, of the model on a sequence and on its reverse complement (referred to as “conjoined post-hoc” in [44]). This asks for more computation at inference time but seems to give the best results overall, as explained in the Abstract of [44]. In fact, the idea to average predictions of multiple models (which in this case is a single non-equivariant model, given two different inputs), is a well-known technique to boost the performance of deep learning models, at the cost of more computation at test time; as one of many examples, Bileschi et al (https://www.biorxiv.org/content/10.1101/626507v4.full.pdf ) show how an ensemble of CNNs outperforms a single CNN on a protein annotation task.
> > >
> > > Since our work is specifically dedicated to the analysis of individual networks (which can be “standard” non-equivariant, or equivariant using either the existing RCPS architecture or the new ones we describe), we did not investigate the orthogonal benefits of model averaging if one is willing to combine several predictions at test time and to pay the corresponding price in computation time (e.g., combining several equivariant networks trained with different random seeds vs combining the predictions of a non-equivariant network on a sequence and its reverse complement). We believe this is a nice research direction for future work, and will be happy to clarify more explicitly this discussion in the final version so that readers do not get confused with what we claim (see also detailed explanations and performance in the response to the previous question)
> > >
> > > Best,
> > >
> > > The authors

---

### Official Review · Reviewer_9RaF · 2021-07-16

**Rating:** 7
**Confidence:** 2

**Summary:**

In this paper, the authors delve deeper into the theoretical foundations of Reverse-complement equivariant convolutions for application to DNA sequences. Prior work has been mainly empirical and this helps to establish a general theoretical framework, thereby encompassing existing RC-equivariant-based models. By identifying this general theory, they were able to propose additional classes of RC equivariant models. They then compare the performance of standard models, existing models, and those proposed as a result from the theory on two regulatory genomics prediction tasks -- binary classification and profile prediction of TF ChIP-seq data. They show that for binary classification, their RC-models perform better than standard networks, but were largely similar for profile prediction.


**Ethical Concerns:**

There are no ethical concerns.

**Limitations And Societal Impact:**

The authors have addressed some limitations when discussing the "disappointing performance" on profile prediction tasks. But they could do a better job of discussing the limitations of their approach to achieve reverse-complement invariance (a stated desired property), but merging methods have limitations at the moment (see commends in main review). There is no mention of negative societal impact of their work as it is unclear how this could be used in a negative way -- so I don't think forcing an arbitrary statement in there would make sense.

**Main Review:**

The work is very interesting and timely. It could open the doors to new types of architectures that incorporate biological priors for genomic data analysis. Such models may lead to better generalization properties, though the evidence here was not as compelling. The main results are theoretical. Although this isn't the first time RC-equivariance was framed using group theory, this paper formalizes it in general terms, which enables them to identify new classes of RC-equivariant models. This is why I believe this work is original and significant. There are a few drawbacks that I outline below. They are mainly about clarity and the very low number of experiments to showcase performance gains and the benefits of RC-equivariant models.

- Disclaimer: I don't have a strong handle on the theoretical aspects of this work, so my evaluation will be mainly on the experiments and results. So, there could be some aspects of it that could be a little more clear. Perhaps a visualization of the theoretical results with a toy example could help understand the different situations (P,a,a) vs (P,a,b) for instance.

- In the text, "DNA sequence classification should therefore be RC-invariant, which calls for RC-equivariant architectures."  RC parameter sharing creates equivariance across reverse-complement transformations in the representations . It does not necessarily mean that if the input sequence were to undergo a RC transformation, then the network predictions will be the same, which is RC-invariance. A strategy for merging the representations is necessary. However, it's not crystal clear to me what is being done because the details of the architecture are in another paper Ref [44], which itself has the information spread between appendix and a few figures. It would be much clearer to present the architectures in detail within the paper.
	a. It seems like the merging mechanisms employed here is to take the sum/mean across the RC feature maps. Doing so indeed ensures RC-invariance. Stating this more clearly could help resolve some of my confusion.
	b. Another  key question I have about RCPS is how are the convolution layers employed? Is it 3 conv layers for each strand (but using reverse complement filters at each layer)? or is it doing the conv with two sets of tied filters (regular and RC) and merging with a sum of the feature maps within each RC conv layer?
	c. What is the difference between standard and RCPS? does standard contain twice the number of filters? For the same number of filters, RCPS effectively has twice as many filters and hence is more expressive. Could the performance gains (to some extent) be coming from this? If not, a better test would be to use twice the number of filters in standard.

- Comment about RC-invariance:  Taking the sum/mean to merge RC feature maps has the effect of ensuring RC-invariance, but it may not necessarily lead to a desirable outcome for DNA sequence analysis. For instance, if a motif were present in one strand, then it will have a high activation for a given filter scan. The RC filter will have a completely different activation structure on the backwards strand, may just be noise. Indeed, if these two feature maps are merged via a mean, then the result would be invariant to whether the forward or reverse strand are fed to this layer. But the act of merging these feature maps averages down meaningful signal and also introduces noise from the wrong RC strand. Ideally, the results should be taken from one strand that the protein binds to and not the average of both, although information about which strand is not known a priori. On the other hand, RC data augmentation and RC conv layers without merging does not necessarily suffer from this issue -- it learns to detect RC motifs efficiently compared to standard convs. It doesn't enforce model will make the same prediction for different strands (i.e. not RC invariant), but it also doesn't average down information in the "correct" but unknown strand and introduce noise into the merged feature map.

- Comment on rigor: There were only 3 binary tasks (CTCF, MAX, and SPI1) for which most claims are drawn from. This is a very small N considering the ease of accessing more TF ChIP-seq datasets for GM12878 cell lines in ENCODE. CTCF and MAX are notoriously simple datasets, based on the performance, I would suspect SPi1 is also. It would be more compelling if the results hold across more experiments.

- besides performance, is there any other benefit to using RC-equivariant models? It's supposedly more efficient, but that wasn't really shown. Perhaps a comparison of standard models with different number of parameters can be compared to the RC-equivariant models. This way you can show that standard models can achieve similar performance, but require X times the number of parameters?

Minor comments:
- The reduced dataset size experiment is a bit odd. It's unclear what the original dataset size was so how much it got reduced to is questionable. The performance of all models remains quite high. Could the dataset get reduced further? perhaps 500 sequences would provide a better separation? Making a systematic plot with varying dataset sizes (a few different choices) would certainly be more informative here.

- k-mer representation was found to be beneficial in the hyperparameter search. Was it explored and found to be beneficial for standard model also?

- In Sec 3, under binary tasks, it was stated that "MAX TF was a failure because of relatively poor performance of RCPS, but using other equivariant networks..." It doesn't seem to be a failure; it performs better than standard? Indeed, Regular and Irrep perform better for this but failure is a poor description.

- For the profile task, the spearman correlation was used. Could/should pearson r, which is a metric that others have used for profile prediciton tasks, also be shown -- it isn't perfect, but the combination of different metrics could help understand the performance better.

- There wasn't a discussion of RC pooling strategies.  I believe a sum or mean is used here -- it wasn't clearly stated and was referred to in another paper. Previously, an RC max-pooling was also suggested in Ref [4].

- Comment on the comparisons: In a previous paper [44], RC data augmentation worked well compared to standard and RCPS. How does it fare here?

**Time Spent Reviewing:**

5

---

> ### Author Response · Authors · 2021-08-10
> **Answer to Reviewer #2 (continued)**
>
> ...
>
> > Minor comments:
> The reduced dataset size experiment is a bit odd. It's unclear what the original dataset size was so how much it got reduced to is questionable. The performance of all models remains quite high. Could the dataset get reduced further? Perhaps 500 sequences would provide a better separation? Making a systematic plot with varying dataset sizes (a few different choices) would certainly be more informative here.
>
> The original data set size depends on the TF, but in orders of magnitude, involves around 30k positive sequences and 100k negative ones. This systematic plot is an easy addition that will be added in the revised version as we agree it would be an informative plot.
>
> > k-mer representation was found to be beneficial in the hyperparameter search. Was it explored and found to be beneficial for the standard model also?
>
> We have not looked into that but expect it to be beneficial too. Once again though, we do not try to boost performance on the task but rather expand the set of admissible RC-equivariant operations and show the potential impact of this expansion.
>
> > In Sec 3, under binary tasks, it was stated that "MAX TF was a failure because of relatively poor performance of RCPS, but using other equivariant networks..." It doesn't seem to be a failure; it performs better than standard? Indeed, Regular and Irrep perform better for this but failure is a poor description.
>
> We will update this.
>
> > For the profile task, the spearman correlation was used. Could/should pearson r, which is a metric that others have used for profile prediction tasks, also be shown -- it isn't perfect, but the combination of different metrics could help understand the performance better.
>
> We have computed other metrics with consistent results, we can add them in the supplementary.
>
> > There wasn't a discussion of RC pooling strategies. I believe a sum or mean is used here -- it wasn't clearly stated and was referred to in another paper. Previously, an RC max-pooling was also suggested in Ref [4].
>
> This was addressed by a comment above : RC pooling is conducted by averaging in the final step, but as a sum by the convolution operations.
>
> > Comment on the comparisons: In a previous paper [44], RC data augmentation worked well compared to standard and RCPS. How does it fare here?
>
> Zhou et al. [44] have shown that, indeed, reverse complement data augmentation improves the performance of the “Standard” model (at the cost of more computation); however, it is significantly worse than RCPS in performance, even after data augmentation. Since we use the same benchmark, we did not feel the need to include the result to the experiment, but following the reviewer’s remark we agree that this is a result we could add as supplementary figure so that the reader has a clear view of a stronger baseline (we re-ran the “Standard” model with data augmentation with our code and obtained the same results as Zhou et al [44]).
>
>
> We thank again Reviewer #2.
>
> Best regards,
>
> The authors

---

> ### Author Response · Authors · 2021-08-10
> **Answer to Reviewer #2**
>
> We thank Reviewer #2 for his review. We refer to the general answer to reviewers for a more general answer to the reviews. However here are the point by point answers to Reviewer #2 ‘s questions.
>
> > [...] So, there could be some aspects of it that could be a little more clear. Perhaps a visualization of the theoretical results with a toy example could help understand the different situations (P,a,a) vs (P,a,b) for instance.
>
> Indeed we decided to provide a rigorous mathematical exposition of our work, and understand that many readers may not be familiar with the math involved. We agree that adding a figure to give an intuitive understanding of the different equivariant layers we describe could be useful, and will do so in the final version. To help users with the code, we also plan to add more examples in the Github repository.
>
> > In the text, "DNA sequence classification should therefore be RC-invariant, which calls for RC-equivariant architectures." RC parameter sharing creates equivariance across reverse-complement transformations in the representations . It does not necessarily mean that if the input sequence were to undergo a RC transformation, then the network predictions will be the same, which is RC-invariance. A strategy for merging the representations is necessary. However, it's not crystal clear to me what is being done because the details of the architecture are in another paper Ref [44], which itself has the information spread between appendix and a few figures. It would be much clearer to present the architectures in detail within the paper. a. It seems like the merging mechanisms employed here is to take the sum/mean across the RC feature maps. Doing so indeed ensures RC-invariance. Stating this more clearly could help resolve some of my confusion. b. Another key question I have about RCPS is how are the convolution layers employed? Is it 3 conv layers for each strand (but using reverse complement filters at each layer)? or is it doing the conv with two sets of tied filters (regular and RC) and merging with a sum of the feature maps within each RC conv layer?
>
> We agree that more details about the models we implement, and more generally how to create an equivariant deep architecture from the basic blocks we describe, would be useful for the readers. We will add this to the final version. In short, we want to be RC-invariant for classification tasks (when we want the prediction to be the same for a sequence and its reverse complement), and to be equivariant to spatial symmetry for profile tasks (when we want the profile prediction for a sequence to be symmetric to the profile prediction of its RC sequence). To achieve this, we note that an RC group action always flips the spatial dimension of the feature maps. It also has a potential additional effect on each positional encoding (no effect for $a_n$ types, switching the signs for $b_n$ types and permutations for the regular ones). For $a_n$ channels, we perform no aggregation. For the $b_n$ and regular type channels, we use averaging over the encodings, by flipping these dimensions along the spatial dimension and averaging each flipped channel for reg or each negated channel for $b_n$. This results in an $a_n$-type layer, where the sequence representation and its RC-complement are just being flipped in the spatial dimension (for instance $000111$ vs $111000$). This is what is needed for the profile tasks. For the binary task, we also add spatial pooling.
>
> Inside the convolutional layers, for instance to get an $a_n$ dimension from a regular one, the constraint on the parameters makes it so that the corresponding dimensions are summed by the convolution operation.
>
> For a given input sequence, we never use the other strand as input in the computations. We just use a special convolution that has the property of tying the feature maps of one input and the feature map one would have had with its RC-complement. These two feature maps can be computed independently from the input and its RC-complement, but they also can be deduced from one another directly with the group action (similarly to how one deduces a RC-complement from a given sequence). In the end, we build a constrained convolution kernel that has approximately twice as less free parameters for a fixed number of filters, and run the same convolution routine.
>
> > c. What is the difference between standard and RCPS? Does the standard contain twice the number of filters? For the same number of filters, RCPS effectively has twice as many filters and hence is more expressive. Could the performance gains (to some extent) be coming from this? If not, a better test would be to use twice the number of filters in standard.
>
> The standard network is just an unconstrained convolutional network, while the RCPS is an equivariant one using solely regular representations. Thus, its convolutional kernels are constrained and for a fixed number of filters, it has less free parameters. We used the same number of free parameters in our plots, but tried to use more filters with a detrimental effect for the standard model. We also tried using data augmentation, a procedure that makes the training longer but improves performance. A figure similar to the ones found in Zhou et al. [44] showing more standard network performance will be added in the revised version.
>
> > Comment about RC-invariance: Taking the sum/mean to merge RC feature maps has the effect of ensuring RC-invariance, but it may not necessarily lead to a desirable outcome for DNA sequence analysis. For instance, if a motif were present in one strand, then it will have a high activation for a given filter scan. The RC filter will have a completely different activation structure on the backwards strand, may just be noise. Indeed, if these two feature maps are merged via a mean, then the result would be invariant to whether the forward or reverse strand are fed to this layer. But the act of merging these feature maps averages down meaningful signal and also introduces noise from the wrong RC strand. Ideally, the results should be taken from one strand that the protein binds to and not the average of both, although information about which strand is not known a priori. On the other hand, RC data augmentation and RC conv layers without merging does not necessarily suffer from this issue -- it learns to detect RC motifs efficiently compared to standard convs. It doesn't enforce that the model will make the same prediction for different strands (i.e. not RC invariant), but it also doesn't average down information in the "correct" but unknown strand and introduce noise into the merged feature map.
>
> Regarding the aggregation, Zou et al [44] show that the RCPS architecture (which is a particular case of the equivariant architecture we study) is capable or representing the solution learned by a non-equivariant conjoined model which is trained on individual strands using data augmentation, and predicts the label of a sequence at test time by averaging the predictions for the sequence and its reverse complement. This suggests that there is no particular problem of signal loss when considering equivariant architectures compared to standard conv nets when one seeks RC-invariance.
>
> > Comment on rigor: There were only 3 binary tasks (CTCF, MAX, and SPI1) for which most claims are drawn from. This is a very small N considering the ease of accessing more TF ChIP-seq datasets for GM12878 cell lines in ENCODE. CTCF and MAX are notoriously simple datasets, based on the performance, I would suspect SPi1 is also. It would be more compelling if the results hold across more experiments.
>
> Cf general answer. Overall the validation of equivariant models was conducted by several concurrent work and we mainly wish to unify and broaden them. Our focus was thus to show that we empirically improved some performances using our broadened function class.
> We based this validation to follow the one of Zhou et al [44]. Including more TFs is possible, but we wanted to conduct a grid search on the new layers choices and to limit the computational cost of this experiment.
>
> > Besides performance, is there any other benefit to using RC-equivariant models? It's supposedly more efficient, but that wasn't really shown. Perhaps a comparison of standard models with different numbers of parameters can be compared to the RC-equivariant models. This way you can show that standard models can achieve similar performance, but require X times the number of parameters?
>
> We show that the performance gap increases when the data set reduces which is another advantage. Computationally speaking, because the network does not need to be trained with data augmentation, the convergence is faster for the RC model. Overall we could say that while classical models need to either train with enhanced data to get a better performance (yet not as good) or have a degraded performance, equivariant nets are a way to get the best of both worlds by construction.
> As mentioned above, a figure investigating the effect of increasing the number of parameters will be included in the supplemental.

---

### Official Review · Reviewer_ny7k · 2021-07-17

**Rating:** 7
**Confidence:** 3

**Summary:**

The paper provides a mathematical formalization of published equivariant layers (translation and reverse-complement) for DNA sequences and explores other architecture variants of these layers (linear, nonlinear and k-mers).

**Ethical Concerns:**

Not applicable.

**Limitations And Societal Impact:**

Not applicable.

**Main Review:**

The paper is clearly written and well structured.

The main contribution of the work is the theoretical characterization of translation and reverse complement (RC) equivarariant CNN layers for DNA sequences. It is based mainly on the work of Cohen and Welling on Group equivariant CNNs applied in vision tasks [1, 2, 3]. Regarding the RC equivariance for DNA sequences, it has been used and empirically studied previously in other papers [4,  5, 6]. Moreover, the authors investigate, define theoretically and implement new architecture variants of, linear and nonlinear, RC equivariant layers.

### Comments:

1) In this comprehensive work, the theoretical results are thoroughly presented and explained. However, the used math is too advanced and not always easy to follow. The authors could make use of graphic figures to support the different concepts as it was done for image processing in [1, 2].

2) The paper does not clearly discuss the importance of using the equivariant nonlinear layers for DNA sequence processing. Characterizing only the equivariant linear layers would be sufficient for this work if their empirical limitations were thoroughly analyzed and discussed; and nonlinear layers could be addressed in a separate work.

3) Although the authors implemented the studied layers, the empirical study is quite minimal compared to the previous work of [4]. Only two equivariant networks are empirically evaluated from the set of new investigated networks. Moreover, the evaluation of the models is based simply on their performance on two supervised tasks in regulatory genomics. Further investigations are needed such as the assessment of the quality of the internal representation of the equivariant networks [7].

4) The hyperparameters search should be done separately for each task.  The hyperparameters of models used in the profile task were set based on a search performed with binary tasks. A $k > 2$ or a ratio $a/(a+b) \neq 0.75$ could lead to better performance on the profile task.

5) The related work is correctly cited throughout the paper, but there is no dedicated section to clearly position the current work from other related machine learning contributions.

### Minor comments:

1) Set the same y-axis limit for both panels in figures 3 and 4.
3) Binary task, lines 313 and 314: provide AuROC with standard error.
2) A.1, lines 521, 524 and 525: the second $\rho_1(s)$ should be $\rho_{-1}(s)$

### References:
* [1] T.S. Cohen and M. Welling. Group equivariant convolutional networks. 2016.
* [2] T. S. Cohen and M. Welling. Steerable CNNs. 2017.
* [3] T. S. Cohen, M. Geiger, and M. Weiler. A General Theory of Equivariant CNNs on Homo- geneous Spaces. 2019.
* [4] H. Zhou, A. Shrikumar, and A. Kundaje. Towards a better understanding of reverse-complement equivariance for deep learning models in regulatory genomics. 2020.
* [5] R. C. Brown and G. Lunter. An equivariant Bayesian convolutional network predicts recom- bination hotspots and accurately resolves binding motifs. 2019.
* [6] Bartoszewicz, Jakub M., et al. DeePaC: predicting pathogenic potential of novel DNA with reverse-complement neural networks. 2020.
* [7] A. Shrikumar, P. Greenside, and A. Kundaje. Reverse-complement parameter sharing improves deep learning models for genomics. 2017.

**Time Spent Reviewing:**

15

---

> ### Author Response · Authors · 2021-08-10
> **Answer to reviewer #1**
>
> We thank Reviewer #1 for his review. We refer to the general answer to reviewers for a more general answer to the reviews. However here are the point by point answers to Reviewer #1 ‘s questions.
>
> > In this comprehensive work, the theoretical results are thoroughly presented and explained. However, the math used is too advanced and not always easy to follow. The authors could make use of graphic figures to support the different concepts as it was done for image processing in [1, 2].
>
> Indeed we decided to provide a rigorous mathematical exposition of our work, and understand that many readers may not be familiar with the math involved. We agree that adding a figure to give an intuitive understanding of the different equivariant layers we describe could be useful, and will do so in the final version. To help users with the practical use of the equivariant layers we describe, we also plan to add more examples in the Github repository.
>
> > The paper does not clearly discuss the importance of using the equivariant nonlinear layers for DNA sequence processing. Characterizing only the equivariant linear layers would be sufficient for this work if their empirical limitations were thoroughly analyzed and discussed; and nonlinear layers could be addressed in a separate work.
>
> We would like to clarify that any deep equivariant architecture needs nonlinear layers, otherwise the deep network would just be equivalent to a single linear layer; and in addition, the nonlinear layers need to be equivariant, otherwise the deep network would not be equivariant. Hence we believe it is an important contribution of our work to characterize the building blocks (linear and nonlinear layers) that allow building a deep equivariant architecture, especially because only specific nonlinearities are allowed after some specific linear layers.
>
> > Although the authors implemented the studied layers, the empirical study is quite minimal compared to the previous work of [4]. Only two equivariant networks are empirically evaluated from the set of new investigated networks. Moreover, the evaluation of the models is based simply on their performance on two supervised tasks in regulatory genomics. Further investigations are needed such as the assessment of the quality of the internal representation of the equivariant networks [7].
> The hyperparameters search should be done separately for each task. The hyperparameters of models used in the profile task were set based on a search performed with binary tasks. A k>2 or a ratio a/(a+b)≠0.75 could lead to better performance on the profile task.
>
> We agree that our empirical study is not as thorough as [4], mainly because our main contribution is theoretical (characterize all equivariant architectures for DNA) and we only wanted to illustrate empirically that exploring the space of equivariant architectures can be beneficial. In addition, by using the same benchmark as [4], we do not need to re-do all the experimental tests done in that paper on non-equivariant and RCPS-equivariant architectures, and can simply borrow their results for comparison. In our experiments where we explore different equivariant architectures, we do not see that a particular architecture is always the best, and we do take this as an important take-away: optimizing the architecture of the network for each specific task may be important to get good results. We acknowledge that a more thorough architecture and hyper-parameter search would most probably result in enhanced performances on each individual task. Another reason why we did not explore more the hyperparameters is that our grid search made us train 720 networks for about 10 GPU days, and conducting a full grid search for the second task would be much more computationally intensive for a limited benefit.
>
> > The related work is correctly cited throughout the paper, but there is no dedicated section to clearly position the current work from other related machine learning contributions.
>
> We organized the paper without a dedicated section about related work, but instead provided in the introduction a presentation of the related work and our contributions. We propose to add in the last paragraph of the introduction (“our contribution”) a clearer description of the position of our work with respect to existing work, in particular for our results on equivariant nonlinear layers.
>
> We will address the minor comments in the final version.
>
> We thank again Reviewer #1.
>
> Best regards,
>
> The authors.

---

### Decision · Program_Chairs · 2021-09-27

**Decision:**

Accept (Poster)

**Comment:**

This paper addresses a very interesting problem motivated by genomics data analysis. All the 3 reviewers unanimously suggest to accept this paper and the meta reviewer agrees with the reviewers. Thus an acceptance is recommended.